# Brain encoding models based on multimodal transformers can transfer across language and vision

**Jerry Tang**
UT Austin
jerrytang@utexas.edu

**Meng Du**
Intel Labs, UCLA
mengdu@ucla.edu

**Vy A. Vo**
Intel Labs
vy.vo@intel.com

**Vasudev Lal**
Intel Labs
vasudev.lal@intel.com

**Alexander G. Huth**
UT Austin
huth@cs.utexas.edu

## Abstract

Encoding models have been used to assess how the human brain represents concepts in language and vision. While language and vision rely on similar concept representations, current encoding models are typically trained and tested on brain responses to each modality in isolation. Recent advances in multimodal pretraining have produced transformers that can extract aligned representations of concepts in language and vision. In this work, we used representations from multimodal transformers to train encoding models that can transfer across fMRI responses to stories and movies. We found that encoding models trained on brain responses to one modality can successfully predict brain responses to the other modality, particularly in cortical regions that represent conceptual meaning. Further analysis of these encoding models revealed shared semantic dimensions that underlie concept representations in language and vision. Comparing encoding models trained using representations from multimodal and unimodal transformers, we found that multimodal transformers learn more aligned representations of concepts in language and vision. Our results demonstrate how multimodal transformers can provide insights into the brain's capacity for multimodal processing.

## 1 Introduction

Encoding models predict brain responses from quantitative features of the stimuli that elicited them [1]. In recent years, fitting encoding models to data from functional magnetic resonance imaging (fMRI) experiments has become a powerful approach for understanding information processing in the brain. While many studies have shown that encoding models achieve good performance when trained and tested on brain responses to a single stimulus modality, such as language [2–9] or vision [10–15], the human brain is remarkable in its ability to integrate information across multiple modalities. There is growing evidence that this capacity for multimodal processing is supported by aligned cortical representations of the same concepts in different modalities—for instance, hearing "a dog is chasing a cat" and seeing a dog chasing a cat may elicit similar patterns of brain activity [16–21].

In this work, we investigated the alignment between language and visual representations in the brain by training encoding models on fMRI responses to one modality and testing them on fMRI responses to the other modality. Encoding models that successfully transfer across modalities can provide insights into how the two modalities are related [20]. Although previous work has compared language and vision encoding models, human annotations were required to map language and visual stimuli into a shared semantic space [20]. To our knowledge, cross-modality transfer has yet to be

37th Conference on Neural Information Processing Systems (NeurIPS 2023).

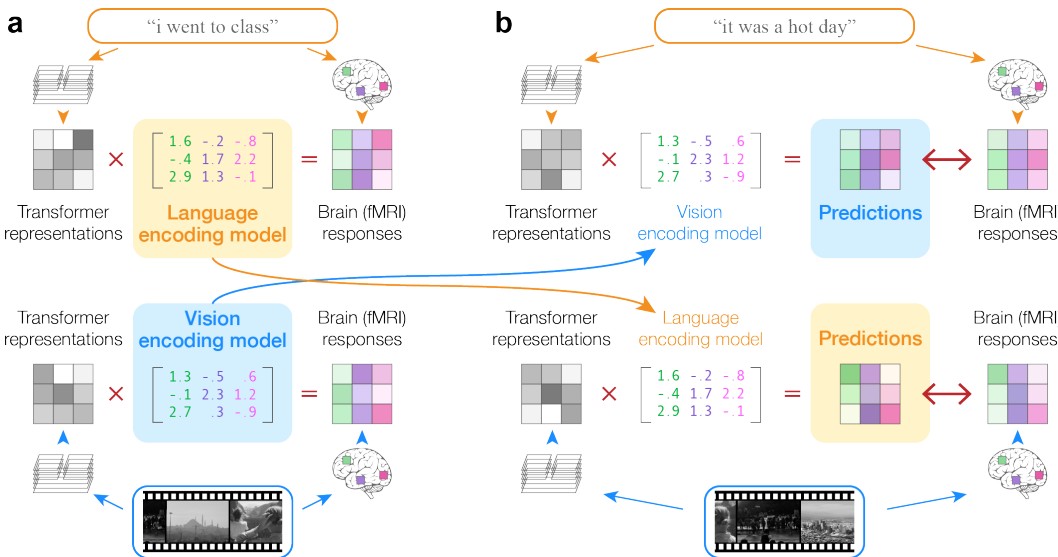

Figure 1: Cross-modality encoding model transfer. (**a**) Multimodal transformers were used to extract features of story and movie stimuli. Language encoding models were fit (using L2-regularized linear regression) to predict story fMRI responses from story stimuli; vision encoding models were fit to predict movie fMRI responses from movie stimuli. (**b**) As the language and vision encoding models share a representational space, language encoding models could be used to predict fMRI responses to movie stimuli, and vision encoding models to predict fMRI responses to story stimuli. Encoding model performance was quantified by the linear correlation between the predicted and the actual response time-courses in each voxel.

demonstrated using encoding models trained on stimulus-computable features that capture the rich connections between language and vision.

One way to extract aligned features of language and visual stimuli is using transformer models trained on multimodal objectives like image-text matching [22–28]. Recent studies have shown that multimodal transformers can model brain responses to language and visual stimuli [29], often outperforming unimodal transformers [30–32], suggesting that multimodal training enables transformers to learn brain-like representations. However, these studies do not assess whether representations from multimodal transformers can be used to train encoding models that transfer across modalities. Since multimodal transformers are trained to process paired language and visual inputs, the representations learned for a concept in language could be correlated with the representations learned for that concept in vision. This alignment between language and visual representations in multimodal transformers could facilitate the transfer of encoding models across modalities, which relies on the alignment between language and visual representations in the brain.

To test this, we used the BridgeTower [28] multimodal transformer to model fMRI responses to naturalistic stories [9] and movies [12]. We separately obtained quantitative features of story and movie stimuli by extracting latent representations from BridgeTower. We estimated language encoding models using story features and story fMRI responses, and vision encoding models using movie features and movie fMRI responses (Figure 1a). We evaluated how well the language encoding models can predict movie fMRI responses from movie features ($story \rightarrow movie$) and how well the vision encoding models can predict story fMRI responses from story features ($movie \rightarrow story$) (Figure 1b). We compared this to how well the language encoding models can predict story fMRI responses from story features ($story \rightarrow story$) and how well the vision encoding models can predict movie fMRI responses from movie features ($movie \rightarrow movie$).

We found that encoding models trained on brain responses to one modality could accurately predict brain responses to the other modality. In many brain regions outside of sensory and motor cortex, $story \rightarrow movie$ performance approached $movie \rightarrow movie$ performance, suggesting that these regions encode highly similar representations of concepts in language and vision. To assess these representations, we performed principal components analysis on the encoding model weights and identified semantic dimensions that are shared between concept representations in language and vision. Finally, we found that cross-modality performance was higher for features extracted from multimodal transformers than for linearly aligned features extracted from unimodal transformers.

Our results characterize how concepts in language and vision are aligned in the brain and demonstrate that multimodal transformers can learn representations that reflect this alignment.

## 2 Multimodal transformers

Multimodal transformers are trained on paired language and visual data to perform self-supervised tasks such as image-text matching. Typically, these models are used to extract representations of paired language and visual input for downstream tasks such as visual question answering. However, since multimodal training objectives may impose some degree of alignment between language and visual tokens for the same concept, these models could also be used to extract aligned representations of language and visual input in isolation. For instance, latent representations extracted from the sentence "a dog is chasing a cat" may be correlated with latent representations extracted from a picture of a dog chasing a cat.

### 2.1 Feature extraction

In this study, we extracted stimulus features using a pretrained BridgeTower model [28]. BridgeTower is a vision-language transformer trained on image-caption pairs from the Conceptual Captions [33], SBU Captions [34], MS COCO Captions [35], and Visual Genome [36] datasets. For each image-caption pair, the caption is processed using a language encoder initialized with pretrained RoBERTa parameters [37] while the image is processed using a vision encoder initialized with pretrained ViT parameters [38]. The early layers of BridgeTower process language and visual tokens independently, while the later layers of BridgeTower are cross-modal layers that process language and visual tokens together. Results are shown for the BridgeTower-Base model; corresponding results for the BridgeTower-Large model are shown in Appendix F.

We used BridgeTower to extract features from the story (Section 3.1) and movie (Section 3.2) stimuli that were used in the fMRI experiments. Each story and movie stimulus was separately processed using BridgeTower by running forward passes with input from the corresponding modality. Hidden representations were extracted from each layer of BridgeTower as the inputs were processed.

For stories, segments of transcripts were provided to the model without accompanying visual inputs. A feature vector was obtained for every word by padding the target word with a context of 20 words both before and after. For movies, single frames were provided to the model without accompanying language inputs. Movies were presented at 15 frames per second, and a feature vector was obtained for every 2-second segment by averaging latent representations across the 30 corresponding frames. Feature extraction was done on a node with 10 Intel Xeon Platinum 8180 CPUs and an Nvidia Quadro RTX 6000 GPU.

### 2.2 Feature alignment

Transformers compute representations for each layer by attending to different combinations of the input tokens. While multimodal training tasks may require models to align language and visual tokens for the same concept, the nature of this alignment depends on the type of attention mechanism used to combine language and visual tokens [39]. BridgeTower uses a co-attention mechanism wherein language and visual tokens are passed through different projection matrices, and query vectors from each modality are only scored against key vectors from the other modality. As a consequence, the language and visual feature spaces extracted from each layer of BridgeTower may differ up to a linear transformation.

To correct for these potential transformations, we used the Flickr30K dataset [40]—which consists of paired captions and images—to estimate linear transformation matrices that explicitly align the language and visual feature spaces extracted from BridgeTower. We first used BridgeTower to separately extract language features of each caption and visual features of each image. We then used L2-regularized linear regression to estimate $image \rightarrow caption$ matrices that predict each language feature from the visual features, and $caption \rightarrow image$ matrices that predict each visual feature from the language features. We evaluated the matrices by measuring how well they align language and visual features for held out samples from the Flickr30K dataset, and found that the matrices achieved significant alignment of language and visual features (Appendix C).

Before using the language encoding model to predict movie fMRI responses, we first used the $image \to caption$ matrix to project the movie features into the language feature space. Similarly, before using the vision encoding model to predict story fMRI responses, we first used the $caption \to image$ matrix to project the story features into the visual feature space.

To show that this explicit feature alignment is only necessary for certain attention mechanisms, we repeated our analyses using a KD-VLP [24] vision-language transformer (Appendix G). KD-VLP uses a merged attention mechanism wherein language and visual tokens are passed through the same projection matrices and query vectors are scored against all key vectors, making explicit feature alignment unnecessary.

## 3  fMRI experiments

We analyzed publicly available fMRI data from five subjects (2 female, 3 male) who participated in a story listening experiment and a movie watching experiment [20]. Previous studies have modeled the story fMRI data as a function of the story stimuli [9, 4, 41] and the movie fMRI data as a function of the movie stimuli [12]. Blood-oxygen level dependent (BOLD) brain signals were recorded using gradient-echo EPI on a a 3T Siemens TIM Trio scanner at the UC Berkeley Brain Imaging Center with a 32-channel volume coil, TR = 2.0045 seconds, TE = 31 ms, flip angle = 70 degrees, voxel size = 2.24 × 2.24 × 4.1 mm (slice thickness = 3.5 mm with 18 percent slice gap), matrix size = 100 × 100, and 30 axial slices. All experiments and subject compensation were approved by the UC Berkeley Committee for the Protection of Human Subjects. Further details about the data are provided in Appendix A.

### 3.1  Story experiment

Stimuli for the story experiment consisted of 10 naturally spoken narrative stories from The Moth Radio Hour totaling just over 2 hours [9]. The stories were presented over Sensimetrics S14 headphones. Subjects were instructed to listen to the stories with their eyes closed. Each story was played during a single fMRI scan.

### 3.2  Movie experiment

Stimuli for the movie experiment consisted of 12 videos totaling 2 hours [12, 14]. Each video was made by concatenating a sequence of 10-20 s clips from movies drawn from the Apple QuickTime HD gallery and YouTube. The videos were presented at 15 frames per second. The videos originally contained dialogue and music, but were presented silently to the subjects. Subjects were instructed to fixate on a dot at the center of the screen. Each video was played during a single fMRI scan.

## 4  Voxelwise encoding models

Our study builds upon the voxelwise modeling framework used in many previous fMRI studies [1]. Voxelwise encoding models learn a mapping from stimuli to the brain responses that they elicit in each individual subject [9]. Brain images recorded at times $t = 1...T$ are given by $y(t) \in \mathbb{R}^m$ where $m$ is the number of voxels in the cerebral cortex. Responses for one subject are represented by the response matrices $Y_{story} \in \mathbb{R}^{T_{story} \times m}$ and $Y_{movie} \in \mathbb{R}^{T_{movie} \times m}$.

The story and movie features were resampled to the fMRI acquisition times using a Lanczos filter. To account for the hemodynamic response, a finite impulse response model with 4 delays (2, 4, 6, and 8 seconds) was applied to the downsampled features. This resulted in the delayed stimulus matrices $X_{story} \in \mathbb{R}^{T_{story} \times 4k}$ and $X_{movie} \in \mathbb{R}^{T_{movie} \times 4k}$ where $k = 768$ is the dimensionality of the BridgeTower features.

We modeled the mapping between stimulus features and brain responses with a linear model $Y = X\beta$. Each column of $\beta$ represents a voxel's linear weights on the $4k$ delayed features. The weights $\beta$ were estimated using L2-regularized linear regression. Regularization parameters were independently selected for each voxel using 50 iterations of a cross-validation procedure.

Voxelwise modeling was done on a workstation with an Intel Core i9-7900X CPU. Further details about voxelwise modeling are provided in Appendix B.

### 4.1 Evaluation

Voxelwise encoding models were evaluated by predicting the response matrices $Y_{test}$ from the stimulus matrices $X_{test}$ for stimuli that were excluded from model estimation. Prediction performance for each voxel was quantified by the linear correlation between the predicted and the actual response time-courses [9, 4, 6, 2].

To quantify $source \rightarrow target$ performance from a source modality to a target modality, we estimated encoding models using all source scans and evaluated prediction performance on each target scan. We averaged linear correlations across the target scans to obtain a score $r_{source \rightarrow target}$ for each voxel. A high $r_{source \rightarrow target}$ score indicates that a voxel's tuning for concepts in the target modality is aligned with its tuning for concepts in the source modality.

We compared this cross-modality performance against the within-modality performance of an encoding model trained on the target modality. To quantify $target \rightarrow target$ performance, we held out each target scan, estimated encoding models using the remaining target scans, and evaluated prediction performance on the held out target scan. We averaged linear correlations across the held out target scans to obtain a score $r_{target \rightarrow target}$ for each voxel.

Many previous studies evaluated encoding models on averaged responses to a single test stimulus [9, 42, 4]. Here we evaluated encoding models on single repetition responses to many test stimuli, as our stimuli vary in semantic content. While our evaluation produces less biased estimates of cross-modality and within-modality performance, the correlation values will be lower than previously reported results due to the lower signal-to-noise ratio of single repetition response data.

We separately identified voxels with statistically significant $story \rightarrow story$ and $movie \rightarrow movie$ within-modality scores. We computed null distributions of the within-modality scores using a blockwise permutation test to account for autocorrelation in the voxel responses (Appendix C).

### 4.2 Layer selection

Separate encoding models were trained using stimulus features extracted from each layer of BridgeTower. We summarized performance across layers by estimating the best layer for each voxel using a bootstrap procedure. For each test scan, we estimated the best layer for each voxel based on mean prediction performance across the remaining test scans. We then used the selected layer for each voxel to compute prediction performance for that voxel on the held out test scan. We used this procedure for all analyses unless noted otherwise.

## 5 Results

We separately estimated language and vision encoding models for each subject. We used these models to compute $r_{story \rightarrow movie}$, $r_{movie \rightarrow story}$, $r_{story \rightarrow story}$, and $r_{movie \rightarrow movie}$ scores for each voxel.

### 5.1 Cross-modality performance

Cross-modality performance was visualized by projecting $r_{story \rightarrow movie}$ and $r_{movie \rightarrow story}$ scores for each voxel in one subject onto a flattened cortical surface (Figure 2a; see Appendix E for other subjects; see Appendix F for BridgeTower-Large model results and Appendix G for KD-VLP model results). We found positive $r_{story \rightarrow movie}$ and $r_{movie \rightarrow story}$ scores in many parietal, temporal, and frontal regions, which have previously been shown to represent the meaning of concepts in language [9] and vision [12]. The high $story \rightarrow movie$ scores and positive (albeit lower) $movie \rightarrow story$ scores suggest that these voxels have similar tuning for the same concepts across modalities [20].

Conversely, we found negative $r_{story \rightarrow movie}$ and $r_{movie \rightarrow story}$ scores in visual cortex. Previous studies have reported that the tuning to perceptual information in visual cortex may be inverted during conceptual processing in the absence of perception [43, 44]. If there is systematically inverted tuning between language and vision, it should be possible to first estimate which voxels would have negative cross-modality scores using separate validation data, and then multiply their weights by $-1$ before computing $r_{story \rightarrow movie}$ and $r_{movie \rightarrow story}$ on test data. We performed this correction using a bootstrap procedure across the test scans. For each test scan, we estimated which voxels have inverted

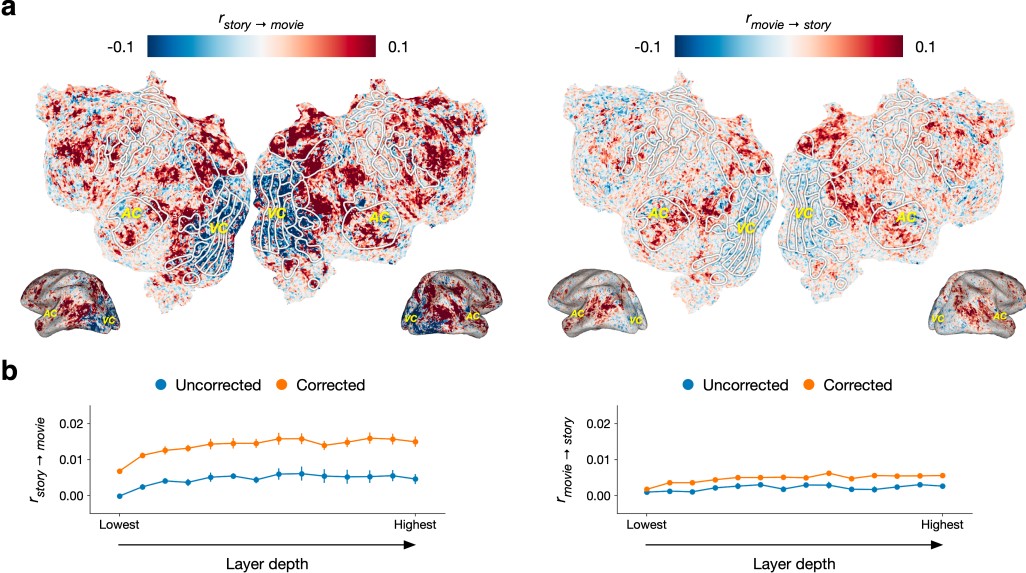

Figure 2: Cross-modality prediction performance. Encoding models estimated on brain responses to one modality were evaluated on brain responses to the other modality. Cross-modality performance is measured by the linear correlation ($r$) between predicted and actual responses. (**a**) $r_{story \rightarrow movie}$ and $r_{movie \rightarrow story}$ scores for each voxel in one subject are displayed on the subject's cortical surface. A voxel appears red if its score is positive, blue if its score is negative, and white if its score is zero. White outlines show regions of interest (ROIs) identified using separate localizer data. AC and VC denote auditory cortex and visual cortex. $r_{story \rightarrow movie}$ and $r_{movie \rightarrow story}$ scores were positive in regions that have previously been found to represent the meaning of concepts in language and vision, but negative in visual cortex. (**b**) Prediction performance for each layer of BridgeTower. Negative scores were corrected by using held out data to fit a one-parameter model for each voxel that predicts whether the encoding model weights should be negated before computing transfer performance. Scores were averaged across voxels and then across subjects to provide unbiased albeit conservatively low summaries of model performance. Error bars indicate standard error of the mean across subjects.

tuning based on the mean prediction performance across the remaining test scans. We then multiplied the weights of these voxels by $-1$ before computing prediction performance on the held out test scan.

Figure 2b shows $story \rightarrow movie$ and $movie \rightarrow story$ performance across cortex before and after this correction. We summarized performance for each layer of BridgeTower by averaging the linear correlations across all cortical voxels and subjects. Averaging across cortical voxels is an unbiased way to compare different encoding models [4], but it produces conservatively low correlation values since many cortical voxels are not involved in processing language or vision. Across layers, the correction significantly improved $story \rightarrow movie$ performance (one-sided paired t-test; $p < 0.05$, $t(4) = 7.5295$, $\overline{r}_{corrected} = 0.0230$, $\overline{r}_{uncorrected} = 0.0053$) and $movie \rightarrow story$ performance (one-sided paired t-test; $p < 0.05$, $t(4) = 6.6356$, $\overline{r}_{corrected} = 0.0097$, $\overline{r}_{uncorrected} = 0.0031$), providing evidence for systematically inverted tuning for the same concepts across modalities.

## 5.2 Comparing cross-modality and within-modality performance

While the previous analysis identified voxels with similar tuning for concepts in language and vision, it did not characterize the extent of this cross-modal similarity. To do this, we next compared cross-modality performance to within-modality performance for each voxel (Figure 3a).

To quantify the amount of information that the language encoding model learns about tuning for movies, we divided $r_{story \rightarrow movie}$ by $r_{movie \rightarrow movie}$ for each voxel. If the movie responses in a voxel are well-predicted by the vision encoding model but poorly predicted by the language encoding model, this value should be low. Conversely, if the movie responses are predicted about as well using both the vision and language encoding models, then this value should be close to 1. In visual cortex, which represents structural features of visual stimuli [14], the language encoding model performed much worse than the vision encoding model at predicting movie responses. In significantly

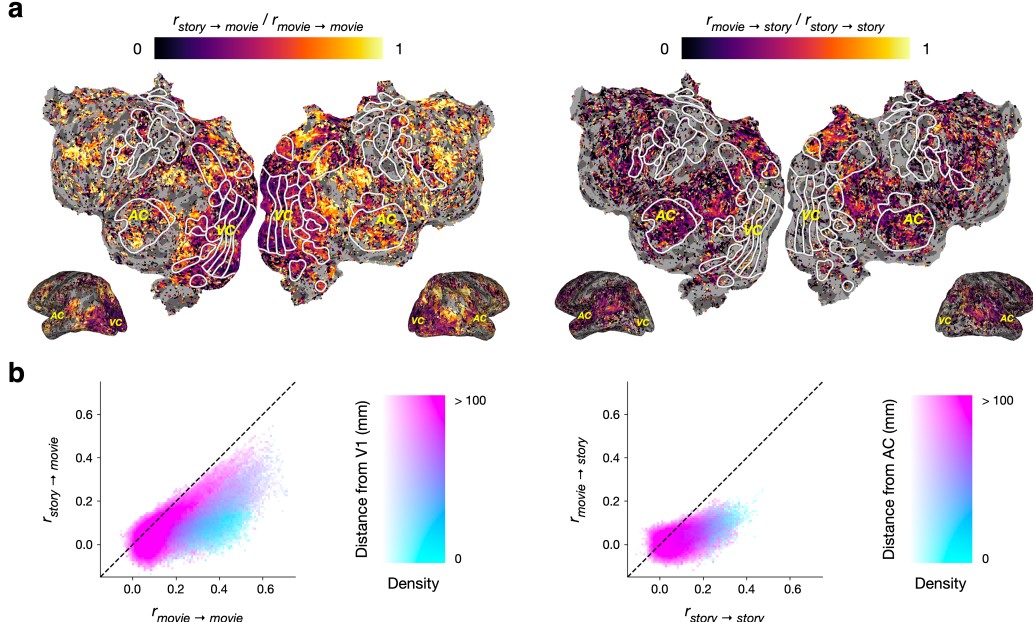

Figure 3: Comparing cross-modality and within-modality prediction performance. Cross-modality scores were compared against within-modality scores in voxels with statistically significant within-modality scores. (**a**) Cross-modality scores and within-modality scores for each voxel in one subject are projected onto the subject's flattened cortical surface. A voxel appears dark if its cross-modality score is much lower than its within-modality score, and bright if its cross-modality score approaches its within-modality score. Only well-predicted voxels under the within-modality model (q(FDR) < 0.05, one-sided permutation test) are shown. (**b**) Histograms compare cross-modality scores to within-modality scores. For movie responses, $r_{story \to movie}$ scores were much lower than $r_{movie \to movie}$ scores near visual cortex but approached $r_{movie \to movie}$ scores in other regions. For story responses, $r_{movie \to story}$ scores were generally much lower than $r_{story \to story}$ scores across cortex.

predicted voxels outside of visual cortex, which represent the meaning of visual stimuli [45, 12, 20], the language encoding model often approached the performance of the vision encoding model (Figure 3b).

Similarly, to quantify the amount of information that the vision encoding model learns about tuning for stories, we divided $r_{movie \to story}$ by $r_{story \to story}$ for each voxel. In auditory cortex, which represents acoustic and articulatory features of language stimuli [42], the vision encoding model performed much worse than the language encoding model at predicting story responses. In some significantly predicted voxels outside of auditory cortex, which have been shown to represent the meaning of language stimuli [42, 9], the vision encoding model performed relatively better, but still did not approach the performance of the language encoding model (Figure 3b).

These results suggest that visual tuning can often be estimated solely based on how a voxel responds to stories, while it is much harder to estimate language tuning solely based on how a voxel responds to movies [46]. One potential confound that could contribute to this asymmetry is that the story stimuli contain both concrete concepts (such as places) and abstract concepts (such as emotions) while the movie stimuli mostly contain concrete concepts. Another potential confound is that the story stimuli contain information at a longer timescale than the movie stimuli, which consist only of 10-20 second clips. To isolate whether the asymmetry in Figure 3 is driven by differences between language and visual representations in the brain, future work could use story and movie stimuli that are matched in terms of semantics and timescale coverage.

## 5.3 Encoding model principal components

The earlier analyses showed that encoding models can readily transfer across modalities, at least in the direction from language to vision. But what kind of information is it that these cross-modal models are capturing? To understand the semantic dimensions that underlie the shared tuning for concepts in

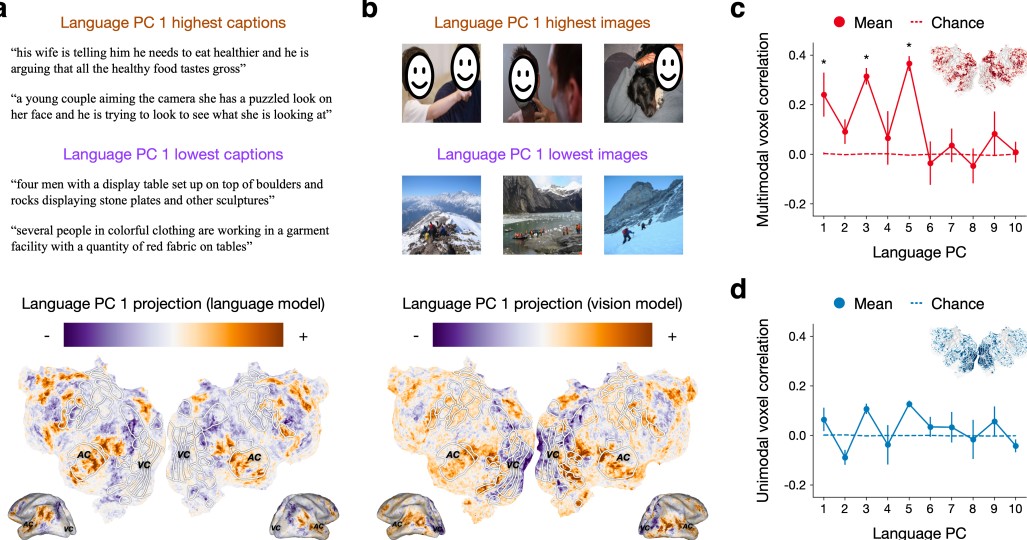

Figure 4: Encoding model principal components. Principal components analysis identified the first 10 principal components (PCs) of language encoding model weights. (**a**) Each caption in Flickr30k was projected onto language PC 1. This PC distinguishes captions that refer to people and social interactions—which are represented in inferior parietal cortex, precuneus, temporal cortex, and frontal cortex—from places and objects—which are represented in superior parietal cortex and middle frontal cortex. (**b**) Each image in Flickr30k was projected onto language PC 1. Here language PC 1 distinguishes images of people—which are represented in EBA, OFA, FFA, inferior parietal cortex, precuneus, temporal cortex, and frontal cortex—from images of places—which are represented in superior parietal cortex and middle frontal cortex. (**c**) In voxels that were well predicted by both the language and the vision encoding models (red on inset flatmap), projections of language and vision encoding model weights were significantly correlated (*) for several language PCs, indicating semantic dimensions that are shared between language and visual representations. (**d**) In voxels that were well predicted by only the language or the vision encoding models (blue on the inset flatmap), projections of language and vision encoding model weights were not significantly correlated for any language PCs. For all results, error bars indicate standard error of the mean across subjects.

language and vision, we next examined the principal components of the language encoding model weights. We first determined the 10,000 best predicted voxels in each subject using a bootstrap procedure (Appendix C). We next averaged the encoding model weights across the 4 delays for each feature to remove temporal information. We finally applied principal components analysis to the averaged encoding model weights, producing 768 orthogonal principal components (PCs) that are ordered by the amount of variance they explain across the voxels. We projected stimulus features onto each PC to interpret the semantic dimension that the PC captures, and we projected encoding model weights onto each PC to assess how the corresponding semantic dimension is represented across cortex. We estimated encoding models using layer 8 of BridgeTower, which has the highest average performance across cortex, and did not correct for negative cross-modality scores.

Projecting Flickr30k caption features onto the first PC of the language encoding model weights (language PC 1), we found that phrases with positive language PC 1 projections tend to refer to people and social interactions, while phrases with negative language PC 1 projections tend to refer to places and objects (Figure 4a). Projecting the language encoding model weights onto language PC 1, we found that voxels with positive projections were mostly located in inferior parietal cortex, precuneus, temporal cortex, and regions of frontal cortex; voxels with negative projections were mostly located in superior parietal cortex and middle frontal cortex. These findings are consistent with previous studies that mapped how different concepts in language are represented across cortex [9, 41].

Since our previous results show that many voxels have shared tuning for concepts in language and vision, the semantic dimensions captured by the language PCs may also underlie the space of visual representations. Projecting Flickr30k image features onto language PC 1, we found that images with positive language PC 1 projections tend to contain people, while images with negative language PC 1 projections tend to contain places such as natural scenes (Figure 4b). Projecting the vision

encoding model weights onto language PC 1, we found similar patterns to the language encoding model projections outside of visual cortex. However, we additionally found voxels with positive projections in visual cortex regions known to represent faces (OFA, FFA) and body parts (EBA) in vision. These results suggest that the semantic dimension captured by language PC 1 is partially shared between language and visual representations.

We next quantified the degree to which each of the top 10 language PCs is shared between language and visual representations. For each PC, we spatially correlated the projections of the language and the vision encoding model weights. We separately computed spatial correlations across multimodal voxels that were well predicted by both the language and vision encoding models—operationalized as the 10,000 voxels with the highest $\min(r_{story \rightarrow story}, r_{movie \rightarrow movie})$—as well as across unimodal voxels that were well predicted by either the language or vision encoding model but not both—operationalized as the 10,000 remaining voxels with the highest $\max(r_{story \rightarrow story}, r_{movie \rightarrow movie})$. The spatial correlations quantify how similarly each population of voxels represents each semantic dimension in language and vision.

We tested the significance of these correlations using a blockwise permutation test (Appendix C). For multimodal voxels (Figure 4c), the projections of the language and the vision encoding model weights were significantly correlated for language PCs 1, 3, and 5 (q(FDR) < 0.05; see Appendix D for further analyses). For unimodal voxels (Figure 4d), the projections of the language and the vision encoding model weights were not significantly correlated for any of the language PCs.

## 5.4 Comparing transfer performance using multimodal and unimodal transformers

Finally, we isolated the effects of multimodal training on cross-modality performance. To provide a unimodal baseline, we estimated language encoding models using RoBERTa [37] and vision encoding models using ViT [38]. Since these unimodal transformers were used to initialize BridgeTower, they provide a baseline for how well language and visual features are aligned prior to multimodal training.

To perform cross-modal transfer with features from the unimodal transformers, we first estimated linear alignment matrices (Section 2.2) on the Flickr30k dataset. We estimated $image \rightarrow caption$ matrices that predict each RoBERTa language feature from the ViT visual features, and $caption \rightarrow image$ matrices that predict each ViT visual feature from the RoBERTa language features. We then used these alignment matrices to evaluate how well a RoBERTa language encoding model can predict movie fMRI responses using ViT movie features, and how well a ViT vision encoding model can predict story fMRI responses using RoBERTa story features.

Across cortex, we found that multimodal features led to significantly higher $story \rightarrow movie$ performance (one-sided paired t-test; $p < 0.05$, $t(4) = 2.1377$, $\overline{r}_{multimodal} = 0.0230$, $\overline{r}_{unimodal} = 0.0219$) and $movie \rightarrow story$ performance (one-sided paired t-test; $p < 0.05$, $t(4) = 5.3746$, $\overline{r}_{multimodal} = 0.0097$, $\overline{r}_{unimodal} = 0.0074$) than unimodal features (Figure 5a). In particular, multimodal features led to higher $story \rightarrow movie$ performance outside of visual cortex, and higher $movie \rightarrow story$ performance outside of auditory cortex (Figure 5b). These results suggest that multimodal training objectives induce the BridgeTower model to learn more complex connections between language and visual representations than a simple linear alignment between modalities.

To isolate the impact of any kind of training on transfer performance, we compared cross-modality performance for a trained BridgeTower model to cross-modality performance for a randomly initialized BridgeTower model. We found that the trained model substantially outperformed the randomly initialized model (Appendix H).

## 6 Discussion

Our study demonstrates that encoding models trained on brain responses to language or visual stimuli can be used to predict brain responses to stimuli in the other modality, indicating similar conceptual representations of language and visual stimuli in the brain [21, 20]. Our analyses identified the regions in which these representations are aligned, as well as the semantic dimensions underlying this alignment. Notably, however, while tuning for concepts in language and vision is positively correlated in most regions outside of visual cortex, it is *negatively* correlated in visual cortex. Understanding the nature of this inverted tuning is an important direction for future work that could provide deeper insights into the relationship between language and vision [43, 44].

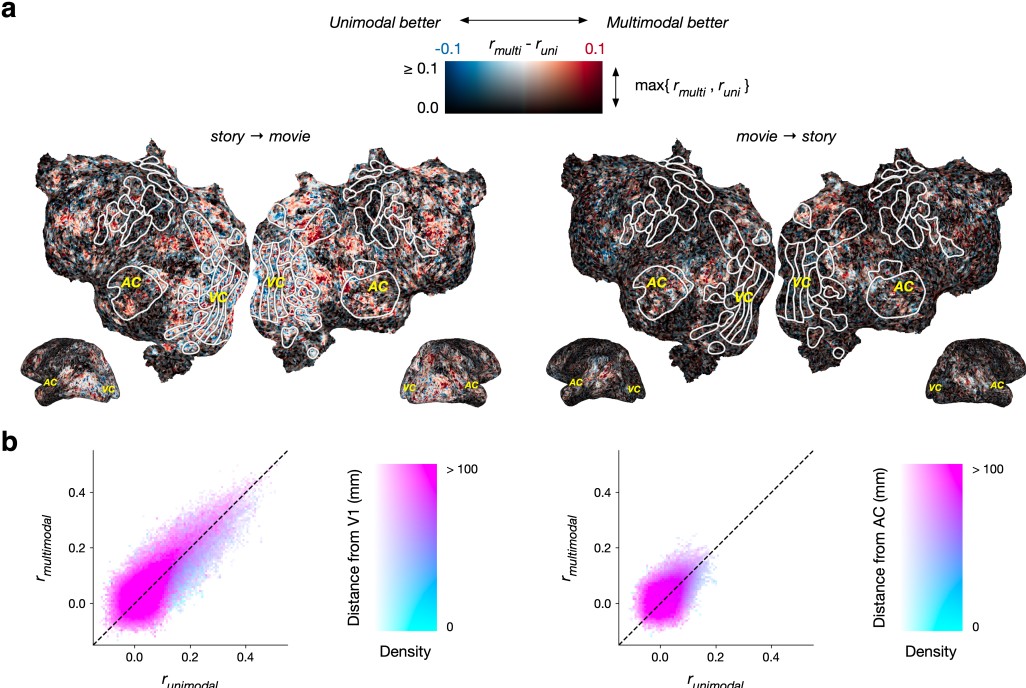

Figure 5: Transfer performance using features from multimodal and unimodal transformers. Cross-modality scores were compared between multimodal encoding models that extract features using BridgeTower and unimodal encoding models that extract features using RoBERTa and ViT. (**a**) The difference between the multimodal score and the unimodal score for each voxel in one subject is projected on the subject's flattened cortical surface. A voxel appears red if it is better predicted by multimodal features, blue if it is better predicted by unimodal features, white if it is well predicted by both, and black if it is well predicted by neither. (**b**) Histograms compare multimodal scores to unimodal scores. For $story \rightarrow movie$ transfer, multimodal features outperform unimodal features in regions outside of visual cortex.

To estimate the cross-modal encoding models, we used the BridgeTower multimodal transformer to extract features of the story and movie stimuli. The successful transfer performance demonstrates that multimodal transformers learn aligned representations of language and visual input. Moreover, stimulus features extracted from multimodal transformers led to better cross-modality performance than linearly aligned stimulus features extracted from unimodal transformers, suggesting that multimodal training tasks enable BridgeTower to learn connections between language and visual concepts that go beyond a simple linear alignment between unimodal representations.

One limitation of our study is the size of the fMRI dataset. Many naturalistic neuroimaging experiments collect a large amount of brain data from a small number of subjects, which allows for the replication of the effects in each individual subject [12, 8, 9, 47, 48]. While our results are consistent across subjects (Appendix E), it would be ideal to replicate these effects in a larger and more diverse population. A second limitation of our study is that the story and movie stimuli are not matched in terms of semantics and timescale coverage, which could lead to asymmetries between $story \rightarrow movie$ and $movie \rightarrow story$ performance. Finally, a third limitation of our study is that there may be relevant stimulus features that are not captured by current multimodal transformers. This could lead us to underestimate the degree of multimodality in the brain. As future multimodal transformers learn to extract increasingly relevant features from story and movie stimuli, new experiments similar to our study might reveal additional brain regions with shared tuning across modalities.

## Acknowledgments and Disclosure of Funding

This work was supported by the National Institute on Deafness and Other Communication Disorders under award number 1R01DC020088-001 (A.G.H.), the Whitehall Foundation (A.G.H.), the Alfred P. Sloan Foundation (A.G.H.) and the Burroughs Wellcome Fund (A.G.H.).

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
