

Figure 6: Coverage of semantic concepts in story and movie stimuli.

# A Dataset

## A.1 Localizers

Regions of interest were identified using functional localizers [12]. Auditory cortex was localized using an auditory localizer consisting of music, speech, and natural sounds. Visual cortex was localized using a retinotopic localizer consisting of clockwise and counterclockwise rotating polar wedges, and expanding and contracting rings.

## A.2 Semantic coverage

To compare the semantic coverage of the story and the movie stimuli, we clustered GloVe vectors [49] into 100 categories using hierarchical agglomerative clustering. We then created 100-dimensional story and movie coverage vectors. The story coverage vector indicates how frequently each of the 100 categories occurs in the stories. The movie coverage vector indicates how frequently each of the 100 categories occurs in the movies, based on WordNet labels assigned to each movie frame [12]. The linear correlation between the story and movie coverage vectors was 0.5, indicating a reasonable degree of overlap.

Figure 6 shows that categories like named locations (e.g. "Georgia", "Texas") occurred more in the stories while categories like nature (e.g. "leaf", "soil") occurred more in the movies. To plot the results on a logarithmic scale, we filtered out categories that did not occur in one of the stimulus modalities. However, such categories typically occurred infrequently in the other modality as well.

# B Voxelwise modeling

In voxelwise modeling, regularized linear regression is used to estimate a set of weights that predict how stimulus features affect the BOLD response in each voxel. We followed the modeling procedure used in previous studies [9].

Stimulus vectors were constructed by extracting features of the story and movie stimuli using multimodal transformers. These vectors were resampled at times corresponding to the fMRI acquisitions using a three-lobe Lanczos filter.

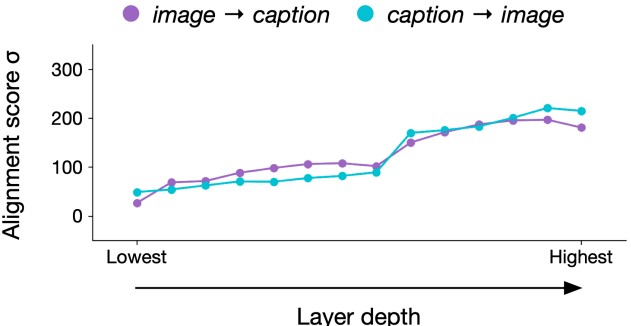

Figure 7: Alignment of BridgeTower-Base features on the Flickr30K dataset.

Linearized finite impulse response (FIR) models were fit to every cortical voxel in each subject's brain. Separate linear temporal filters with four delays ($t-1$, $t-2$, $t-3$ and $t-4$ timepoints) were fit for each of the 768 features, yielding a total of $3,072$ features. With a TR of 2 s, this was accomplished by concatenating the feature vectors from 2 s, 4 s, 6 s and 8 s earlier to predict responses at time $t$. Before doing regression, we first z-scored each feature channel within each stimulus. This was done to match the features to the fMRI responses, which were z-scored within each scan.

The 3,072 weights for each voxel were estimated using L2-regularized linear regression. The regression procedure has a single free parameter that controls the degree of regularization. This regularization coefficient was found for each voxel in each subject by repeating a regression and cross-validation procedure 50 times. In each iteration, approximately a fifth of the timepoints were removed from the model training dataset and reserved for validation. Then, the model weights were estimated on the remaining timepoints for each of 10 possible regularization coefficients (log spaced between 10 and 1,000). These weights were used to predict responses for the reserved timepoints, and then linear correlation was computed between the predicted and the actual responses. For each voxel, the regularization coefficient was chosen as the value that led to the best performance, averaged across bootstraps, on the reserved timepoints.

## C  Significance testing

### C.1  Feature alignment significance test

To test whether a multimodal transformer extracts aligned language features and visual features (Section 2.2), we estimated alignment matrices on $80\%$ of the image-caption pairs in the Flickr30K dataset [40] and evaluated them on the remaining $20\%$. We scored alignment quality by correlating the predicted values and the actual values for each feature across test pairs, and then averaging the correlations across features. Finally, we normalized alignment scores by reporting the number of standard deviations above the mean of a null distribution (denoted by $\sigma$), which we obtained by randomly shuffling the test captions and images.

These results are shown in Figure 7. The $caption \rightarrow image$ matrices had a normalized score of $221.30\sigma$, while the $image \rightarrow caption$ matrices had a normalized score of $197.28\sigma$. Thus in both directions the alignment was roughly 200 standard deviations better than random.

### C.2  Voxel performance significance test

We computed $story \rightarrow story$ and $movie \rightarrow movie$ scores for each voxel by taking the linear correlation between the predicted response time course and the actual response time course. We separately identified voxels with statistically significant $story \rightarrow story$ performance and $movie \rightarrow movie$ performance using a blockwise permutation test.

In each trial, we randomly resampled (with replacement) 10-TR blocks from the voxel's actual response time course, before taking the linear correlation between the predicted response time course and the permuted response time course. Resampling contiguous blocks preserves the auto-correlation

structure of the voxel's responses. Repeating this process for 10,000 trials provided a null distribution of within-modality scores for each voxel. We identified voxels with within-modality scores that were significantly higher than this null distribution than expected by chance (q(FDR) < 0.05).

### C.3 PC correlation significance test

We projected the language and the vision encoding model weights on each principal component (PC) of the language encoding model weights. We assessed the degree to which each PC is shared between language and vision by spatially correlating the projections of the language and the vision encoding model weights. We identified PCs with significant spatial correlations using a blockwise permutation test.

In each trial, we randomly resampled (with replacement) 10-TR blocks from the movie fMRI responses before estimating a null vision encoding model. The projections of the language encoding model weights were correlated with the projections of the null vision encoding model weights. Repeating this process for 1,000 trials provided a null distribution of spatial correlations for each PC. We identified PCs with spatial correlations that were significantly higher than this null distribution than expected by chance (q(FDR) < 0.05).

## D  Principal components

We projected stimulus features onto language PCs 2, 3, 4, and 5 to interpret the semantic dimension that each PC captures, and we projected encoding model weights onto language PCs 2, 3, 4, and 5 to assess how the corresponding semantic dimensions are represented across cortex.

Figure 8 corresponds to Figure 4 in the main text.

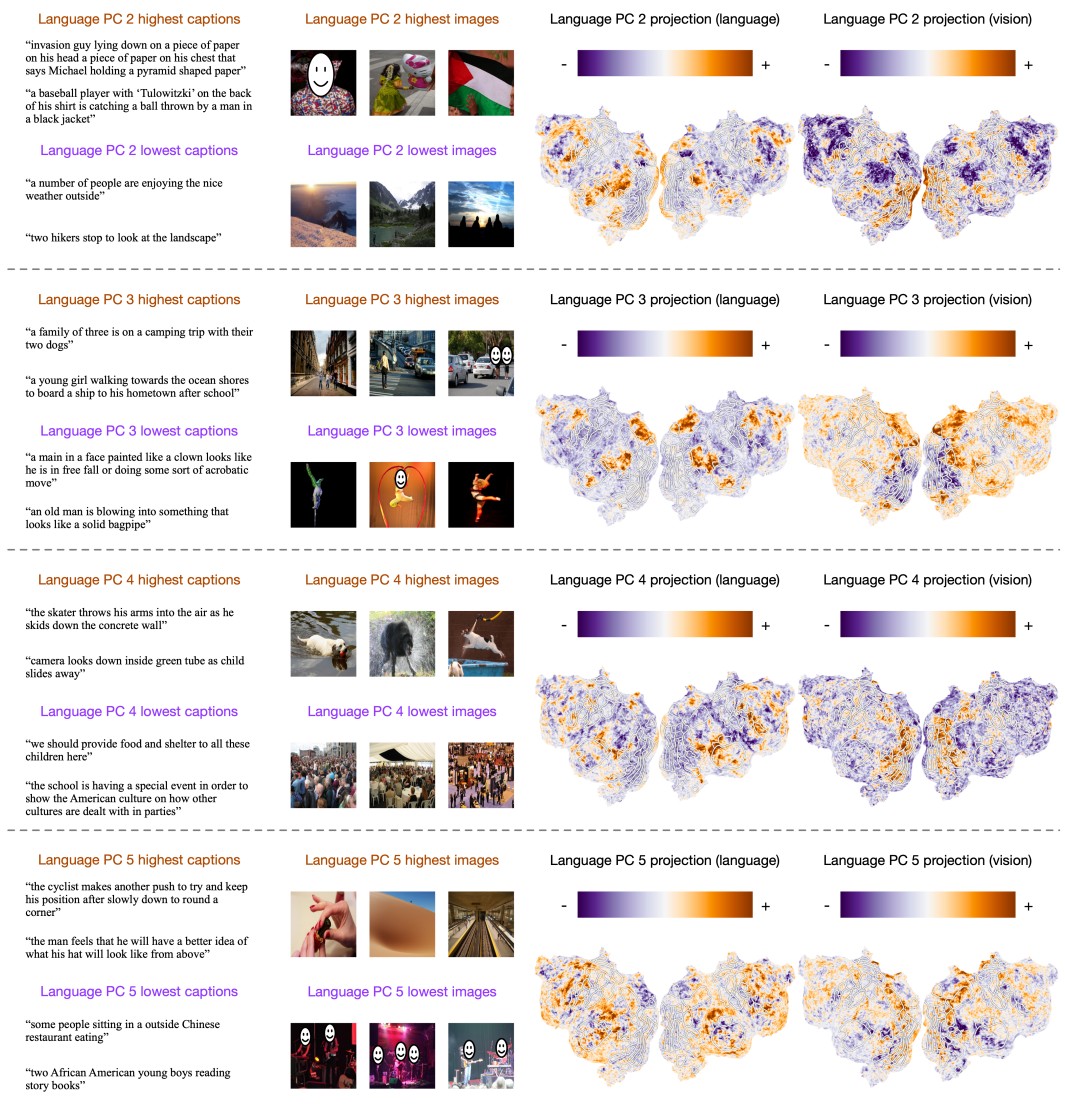

Figure 8: Interpreting language PCs 2, 3, 4, and 5.

# E  Flatmaps

We created flatmaps for subjects S2, S3, S4, S5. We visualized results by projecting scores for each voxel in a subject on the subject's cortical surface. We found similar results across subjects.

Figure 9 corresponds to Figure 2 in the main text. Figure 10 corresponds to Figure 3 in the main text. Figure 11 corresponds to Figure 4 in the main text. Figure 12 corresponds to Figure 5 in the main text.

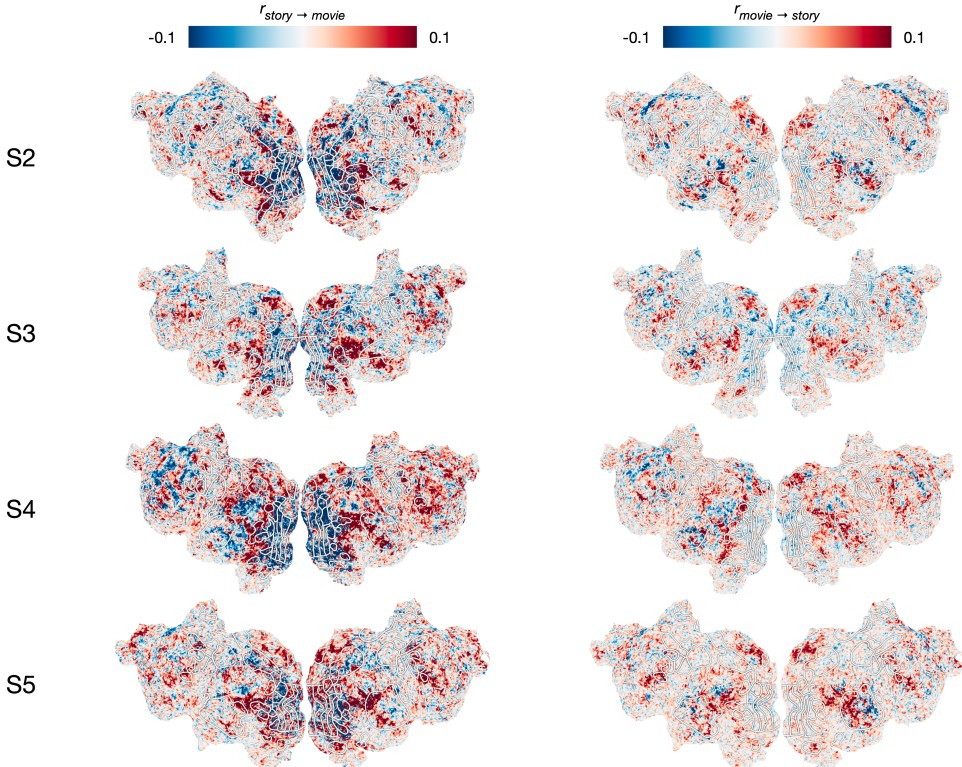

Figure 9: Cross-modality prediction performance for subjects S2, S3, S4, and S5.

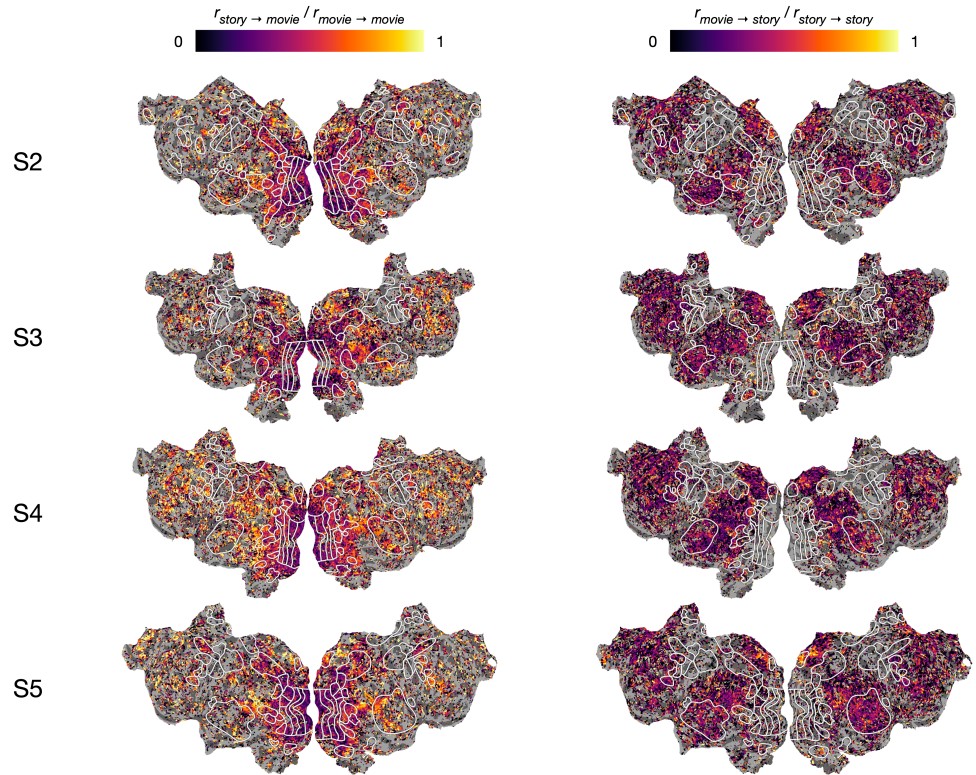

Figure 10: Comparing cross-modality and within-modality prediction performance for subjects S2, S3, S4, and S5.

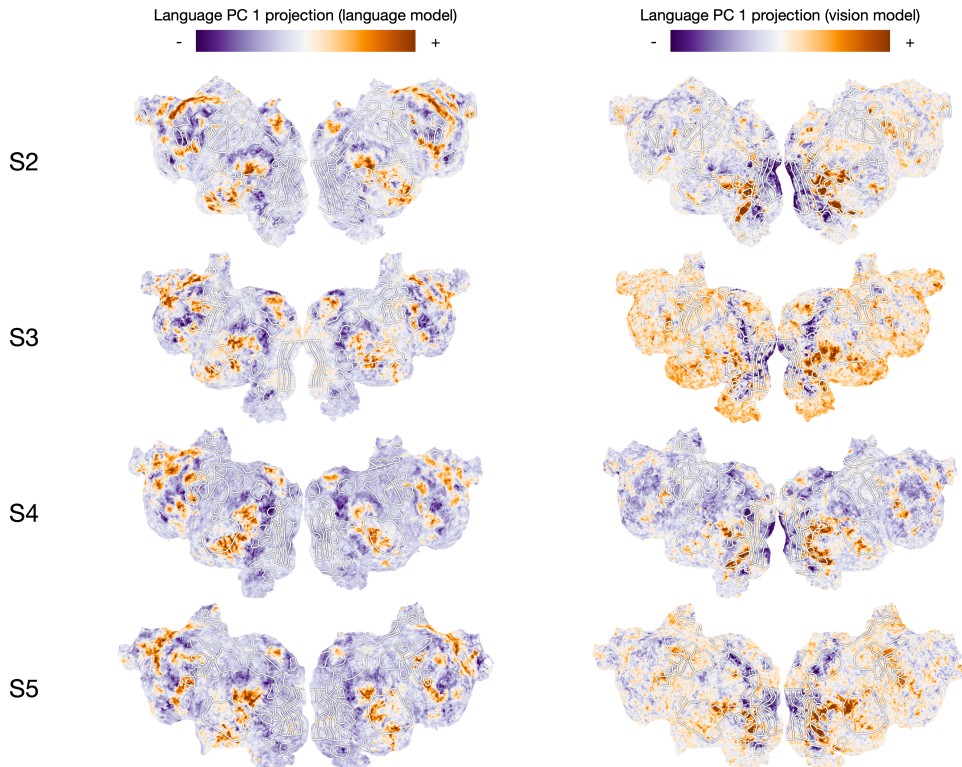

Figure 11: Language PC 1 projections for subjects S2, S3, S4, and S5.

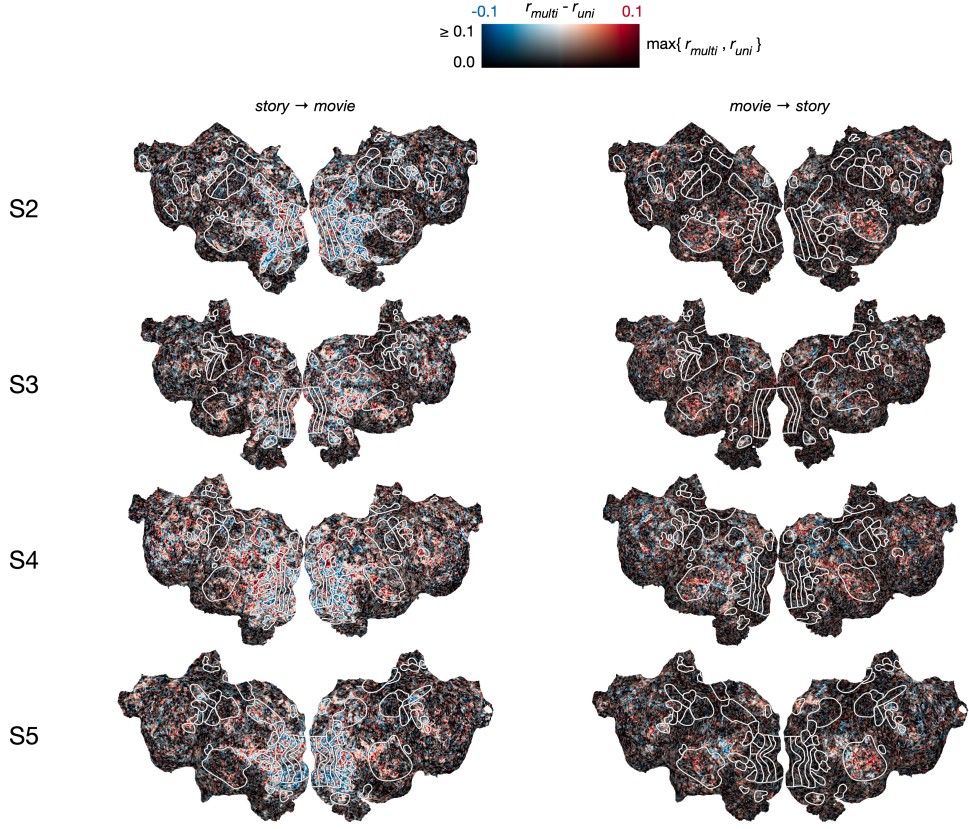

Figure 12: Multimodal and unimodal transfer performance for subjects S2, S3, S4, and S5.

## F BridgeTower-Large

We repeated the analyses in the main text using BridgeTower-Large to extract stimulus features instead of BridgeTower-Base [28]. For unimodal baselines, we used the pretrained RoBERTa-Large [37] and ViT-Large [38] transformers used to initialize BridgeTower-Large. We found similar results for BridgeTower-Base and BridgeTower-Large.

Figure 13 corresponds to Figure 2 in the main text. Figure 14 corresponds to Figure 3 in the main text. Figure 15 corresponds to Figure 4 in the main text. Figure 16 corresponds to Figure 5 in the main text.

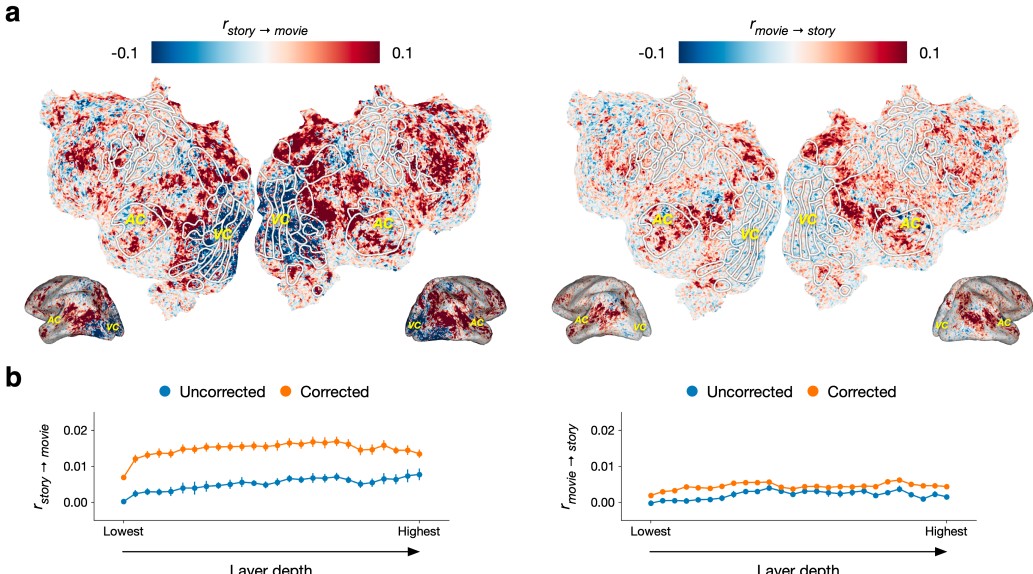

Figure 13: Cross-modality prediction performance for BridgeTower-Large.

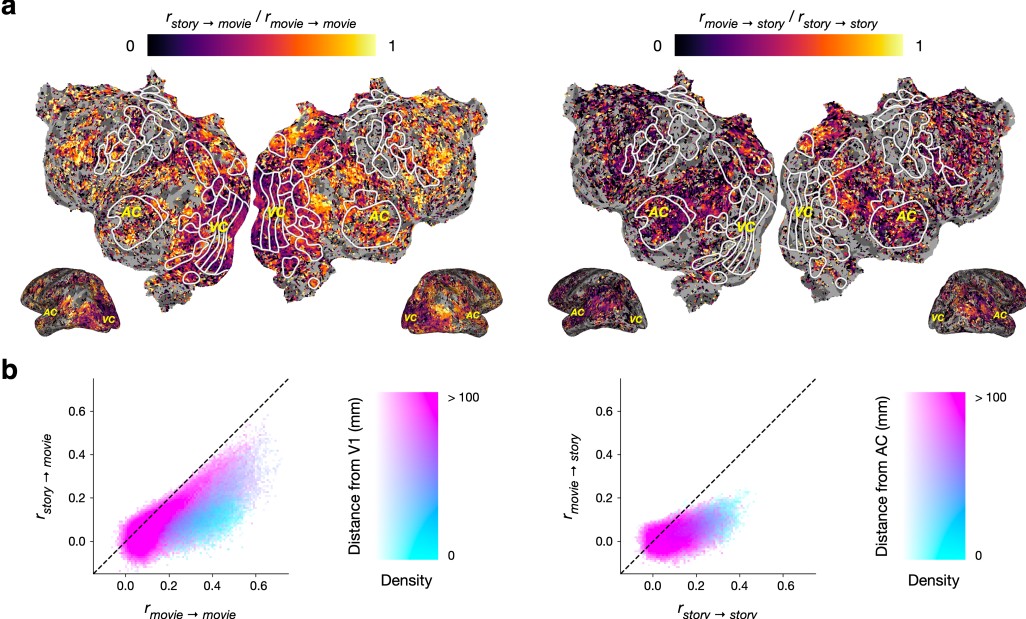

Figure 14: Comparing cross-modality and within-modality for BridgeTower-Large.

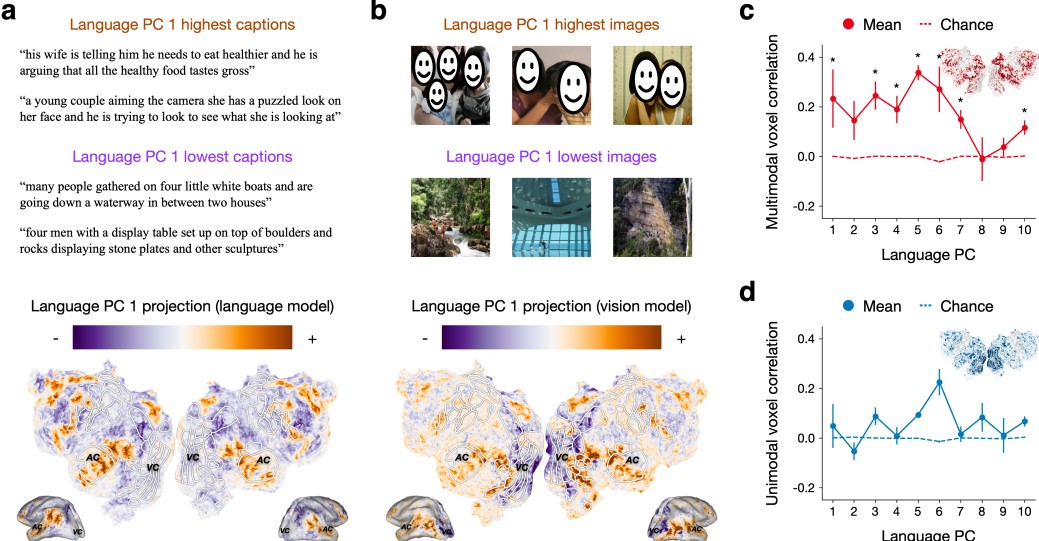

Figure 15: Encoding model principal components for BridgeTower-Large.

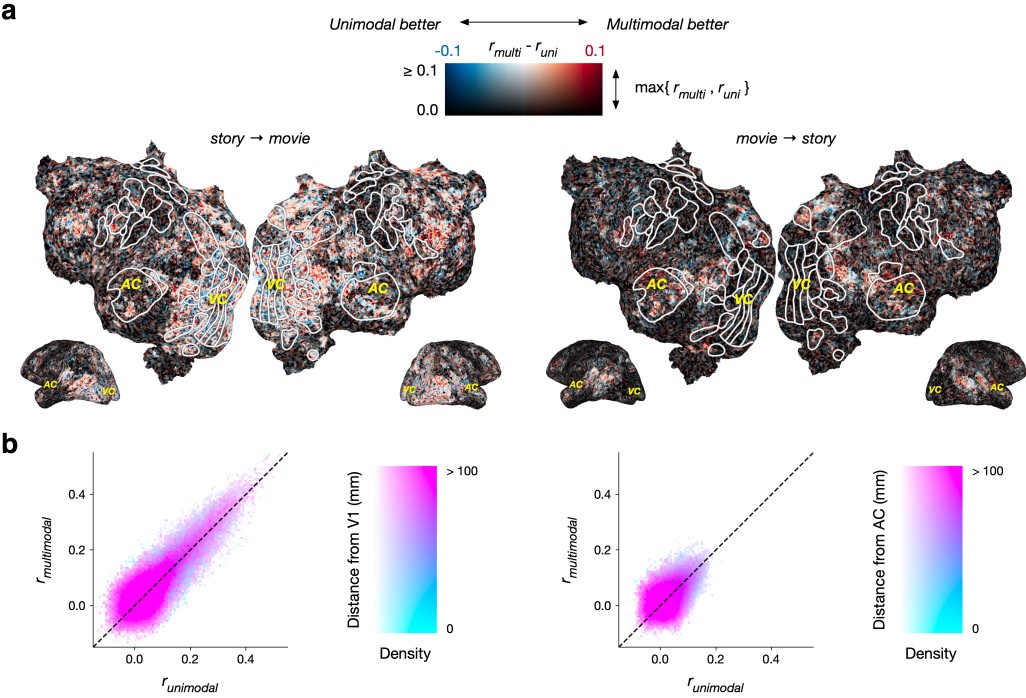

Figure 16: Multimodal and unimodal transfer performance for BridgeTower-Large.

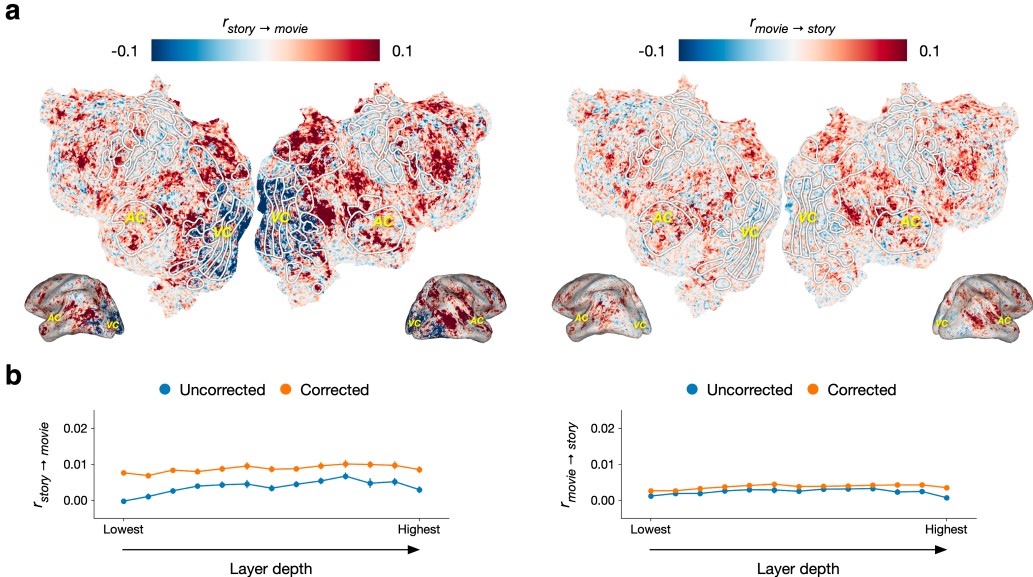

Figure 17: Cross-modality prediction performance for KD-VLP.

## G  KD-VLP

We assessed cross-modality performance when KD-VLP [24] was used to extract stimulus features instead of BridgeTower [28]. KD-VLP is a single-stream multimodal transformer with a merged attention mechanism, whereas BridgeTower is a dual-stream multimodal transformer with a co-attention mechanism [39]. We found similar results for BridgeTower and KD-VLP, demonstrating that our results are consistent across model architectures. For KD-VLP we did not align the language and visual features using $image \rightarrow caption$ and $caption \rightarrow image$ matrices, demonstrating that explicit feature alignment is only necessary for certain model architectures.

Figure 17 corresponds to Figure 2 in the main text.

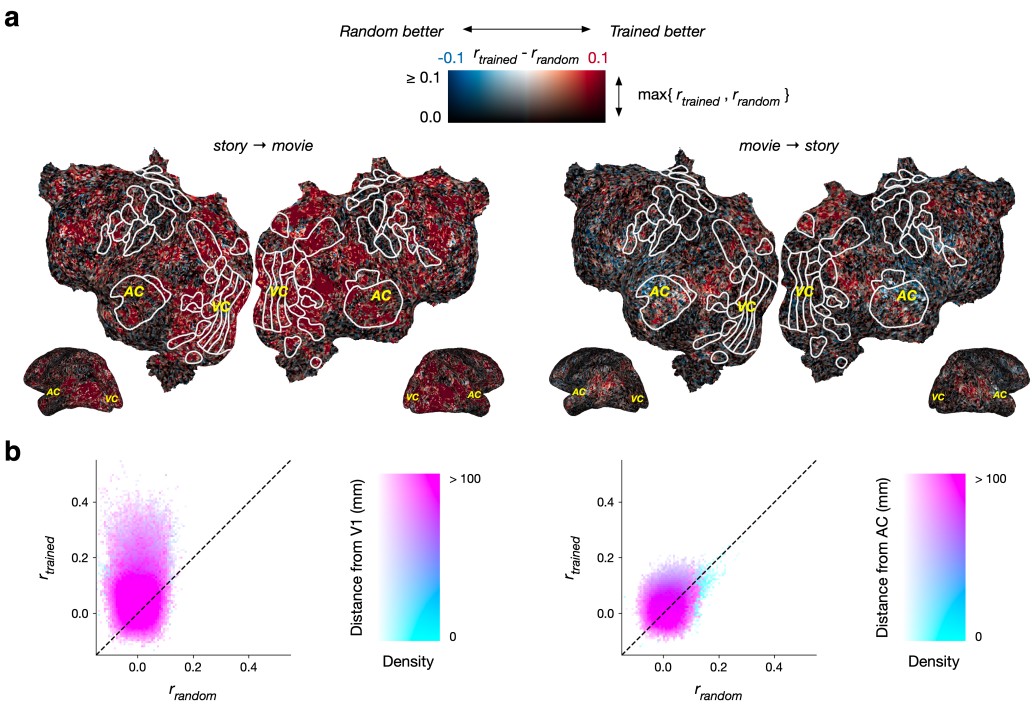

Figure 18: Transfer performance for trained and randomly initialized BridgeTower-Base.

## H    Randomly initialized BridgeTower-Base

We compared cross-modality performance between a trained BridgeTower-Base model and a randomly initialized BridgeTower-Base model. We found that the trained model subtantially outperformed the randomly initialized model, demonstrating the effect of training on cross-modality performance.

Figure 18 corresponds to Figure 5 in the main text.