# OpenReview forum: "Brain encoding models based on multimodal transformers can transfer across language and vision"
_NeurIPS.cc/2023/Conference — NeurIPS 2023 poster_

### Official Review · Reviewer_Ebp3 · 2023-07-03

**Soundness:** 3 good
**Presentation:** 3 good
**Contribution:** 3 good
**Rating:** 8
**Confidence:** 5

**Summary:**

There is a fastly growing literature analyzing how deep learning models' representations have predictive power over fMRI brain measurements. These papers typically train a regressor model (typically a linear regressor) to predict the fMRI measurements from the neural models' representations. Current encoding models are typically trained and tested on fMRI brain activity to each modality (i.e. language or visual or auditory)  in isolation. They then evaluate predictive power by measuring the correlation between predictions and actual measurements.

This paper contributes to that literature by using representations from the multimodal Transformer (BridgeTower) model that can extract aligned representations of concepts in language and vision before using it in the above pipeline. Since language and vision rely on similar concept representations, authors investigated how well the multimodal Transformer representations can transfer across fMRI responses to stories and movies. By comparing the correlation between actual fMRI and predictive fMRI from multimodal and unimodal representations, the authors draw conclusions that multimodal transformers learn more aligned representations of concepts in language and vision. Finally, the authors perform a cross-modal experiment in which how well the language encoding models can predict movie-fMRI responses from narrative story features (story → movie) and how well the vision encoding models can predict narrative story-fMRI responses from movie features (movie → story).

**Strengths:**

The paper contains the following key contributions:
* The novelty of this work: using a multimodal Transformer as the encoder architecture to extract the aligned concept representations for narrative stories and movies to model fMRI responses to naturalistic stories and movies, respectively.
* Encoding models trained on brain responses to one modality could accurately predict brain responses to the other modality.
* Cross-modality performance was higher for features extracted from multimodal transformers than for linearly aligned features extracted from unimodal transformers.

Originality:
* The idea of how concepts in language and vision are aligned in the brain and demonstrated that alignment using multimodal transformers representations.
* Mapping story features -> movie fMRI and movie -> story fMRI is interesting.

Clarity: The paper is written well. The information provided in the submission is sufficient to reproduce the results.

Significance: The idea of using a multimodal Transformer encoder model to extract aligned language-visual concept representations to predict fMRI responses is exciting. The paper has novel cognitive insights about language and visual ROIs; clear experimental evaluation, and adequate related work.

Quality: The paper supports its claims with enough details. The paper is well-written and easy to follow.

**Weaknesses:**

* Since the narrative stories used in the story experiment are completely different from short movie clips, how much is the overlap percentage of concepts in both modalities?
* This problem is more interesting if authors can use the same information for both listening and watching a movie (i.e. the same subject is watching a movie and listening to the same story). Similar to the following work: The same subject was reading and listening same narrative story.  Deniz F., Nunez-Elizalde A.O., Huth A. G., Gallant J. L., The representation of semantic information across human cerebral cortex during listening versus reading is invariant to stimulus modality . Journal of Neuroscience
* It could be interesting if authors could use captions for the videos and compare the performance of story-movie and captions-movie. Similarly, stories-captions and captions-stories.
* Which regions have a higher estimated noise ceiling in listening to and watching a movie on the same subject?
* What is the estimated noise ceiling when using the subject's story voxel to predict the subject movie voxels?
* The authors missed some recent works that studied multimodal Transformers for brain encoding.
Authors missed the following papers:
INTERPRETING MULTIMODAL VIDEO TRANSFORMERS USING BRAIN RECORDINGS, Dota et al. 2023, https://openreview.net/pdf?id=p-vL3rmYoqh
Visio-Linguistic Brain Encoding, Oota et al. 2022, https://aclanthology.org/2022.coling-1.11.pdf


**Questions:**

* Did the authors fine-tune the BridgeTown multimodal Transformer on the current dataset?
* Is there any effect on encoding model performance in which adding noise to one modality features and predicting the other modality fMRI response?
* What if we initialize the model with random weights to perform neural encoding?
* It could be interesting, if authors can test encoding performance for both single-stream and dual-stream multimodal Transformer models?

**Limitations:**

The authors have discussed several limitations and made future directions for the research community.

---

> ### Author Rebuttal · Authors · 2023-08-09
>
> Thank you for reviewing our paper and making these suggestions. We have addressed your concerns in order below:
>
> >Since the narrative stories used in the story experiment are completely different from short movie clips…
>
> Thank you for raising this point. We will describe the concepts contained in the story and movie stimuli in the Appendix of the final paper.
>
> To approximate the overlap in concepts, we clustered GloVe vectors into 100 semantic categories. We then created 100-dimensional story and movie category vectors. The story category vector indicates how frequently each of the 100 categories occurs in the stories. The movie category vector indicates how frequently each of the 100 categories occurs in the movies, based on WordNet labels assigned to each movie frame [1]. The linear correlation between the story and movie category vectors was 0.5, indicating a reasonable degree of overlap.
>
> We plotted the category vectors in Figure R2, which will be included in the Appendix of the final paper. Categories like location proper nouns (e.g. “Brooklyn”, “Ithaca”) occurred more in the stories. Categories like nature words (e.g. “bloom”, “soil”) occurred more in the movies. To plot the results on a log scale, we filtered out categories which did not occur in at least one of the stimulus modalities. However, such categories typically occurred infrequently in the other modality as well.
>
> >This problem is more interesting if authors can use the same information for both listening and watching a movie…
>
> We agree that using stimuli that are more closely matched in terms of semantics and timescales is an important direction for future work (line 217). We will emphasize this future direction in the discussion of the final paper.
>
> >It could be interesting if authors could use captions for the videos and compare the performance of story-movie and captions-movie…
>
> Since the movies were constructed by concatenating a sequence of 10-20 s video clips, there are unfortunately no high-quality captions readily available for this comparison. However, we agree that this is an interesting direction for future work. Previous studies have found that encoding models trained on caption features could be used to predict brain responses to images, albeit not as accurately as encoding models trained on image features (see [5]).
>
> >Which regions have a higher estimated noise ceiling in listening to and watching a movie…
>
> We evaluated our encoding models on single-trial responses, rather than responses to multiple repeats of the same stimulus. While evaluating on single-trial responses to a large number of stimuli provides high semantic coverage, there is no way to estimate a noise ceiling using single-trial responses.
>
> The dataset that we used [3] also includes responses to multiple repeats of a test stimulus. However, the dataset only provides the averaged responses across repeats, whereas noise ceiling estimation requires having the responses to each individual repeat.
>
> >What is the estimated noise ceiling when using the subject's story voxel to predict the subject movie voxels?
>
> The noise ceiling is a function of the test data rather than the encoding model, so the $source→target$ noise ceiling will be the same as the $target→target$ noise ceiling. As we expect cross-modality performance to be lower than within-modality performance, $r_{target→target}$ provides an approximate lower bound on the noise ceiling for $r_{source→target}$ (Figure 3).
>
> >The authors missed some recent works that studied multimodal Transformers for brain encoding…
>
> Thank you for linking to these papers. We agree that these studies are highly relevant, and we will reference them in the final paper.
>
> >Did the authors fine-tune the BridgeTown multimodal Transformer on the current dataset?
>
> No, we did not. We used BridgeTower models that were pretrained on separate datasets (please refer to Section 2.1 for details).
>
> >Is there any effect on encoding model performance in which adding noise to one modality features and predicting the other modality fMRI response?
>
> To test this, we added Gaussian random noise to the stimulus features. For $story→movie$ transfer, prediction performance was robust to noise in the $story$ features. For $movie→story$ transfer, prediction performance slightly increased when noise was added to the movie features, which may result from the noise affecting negative as well as positive correlations. These results are shown in Figure R3.
>
> >What if we initialize the model with random weights to perform neural encoding?
>
> Previous studies have found that within-modality performance for randomly initialized neural networks is above chance, but below that of trained neural networks [13]. After fitting alignment matrices, we expect cross-modality performance to be above chance, but below that of trained unimodal neural networks (Section 5.4) and trained multimodal neural networks. We are currently waiting on results of this experiment, which we will include in the final paper. That said, we believe that the trained unimodal neural networks provide the most direct control for evaluating the effects of multimodal training.
>
> >It could be interesting, if authors can test encoding performance for both single-stream and dual-stream multimodal Transformer models?
>
> To test this, we performed many of the same analyses using KD-VLP [26] which is a single-stream multimodal transformer. These results are shown in Figure R4. We found very similar patterns of transfer performance across cortex. Across voxels and subjects, KD-VLP performance ($r_{story→movie} = 0.17$, $r_{movie→story} = 0.08$) was slightly worse than BridgeTower performance ($r_{story→movie} = 0.23$, $r_{movie→story} = 0.10$). However, these differences could be due to training data and training tasks, as well as model architectures. We will include this analysis in the Appendix of the final paper.

---

> > ### Comment · Reviewer_Ebp3 · 2023-08-15
> > **Thanks for the rebuttal**
> >
> > Dear authors,
> >
> > Thanks for the rebuttal.
> >
> > I appreciate the time and effort, especially in addressing all the questions. However, I still retain some reservations regarding the following inquiry:
> >
> > "What is the estimated noise ceiling when using the subject's story voxel to predict the subject movie voxels?"
> >
> > In light of Schrimpf et al.'s 2020 paper, where they managed to estimate the noise ceiling even with single-trial stimuli, I find it intriguing if the authors could elaborate more on their subsequent answer and potentially draw comparisons to the work conducted by Schrimpf et al. in 2020.
> >
> > Schrimpf et al.'s 2020, The neural architecture of language: Integrative modeling converges on predictive processing, PNAS-2020
> >
> > Your response stated: "We evaluated our encoding models on single-trial responses, rather than responses to multiple repeats of the same stimulus. While evaluating on single-trial responses to a large number of stimuli provides high semantic coverage, there is no way to estimate a noise ceiling using single-trial responses."

---

> > > ### Author Response · Authors · 2023-08-16
> > >
> > > Thanks for the response! The Schrimpf et al. paper uses data from other subjects to estimate noise ceilings for a held-out subject.  As we understand it, this inter-subject approach essentially treats the responses from the other subjects as repeats of the responses from the held-out subject.
> > >
> > > In our study, we computed model performance for each individual voxel. To estimate the Schrimpf et al. noise ceiling for a voxel in a held-out subject, we would need to find an analog of that voxel in every other subject. Since the subjects’ brains differ in anatomical and functional organization, this requires aligning voxels across subjects using anatomical or functional approaches.
> > >
> > > Anatomical approaches are imprecise due to the variability of the semantic system (Huth et al. 2016). Concretely, we could project each subject’s responses into some shared space, and estimate the noise ceiling for the held-out subject’s voxel at coordinate (x, y, z) using other subjects’ voxels at coordinate (x, y, z). However, these voxels may not share any semantic tuning because fine-grained functional properties are not precisely coupled to anatomy. Computing a noise ceiling for a target voxel using unrelated voxels can lead to a low noise ceiling that inflates the results of our study.
> > >
> > > Functional approaches typically rely on regions of interest (ROIs) defined using an independent localizer task. However, as mentioned in our response to Reviewer C9Qg, we do not believe that there is an effective multimodal localizer that would select the voxels that we are interested in evaluating. Functional approaches for voxel-level alignment are less common, and we believe that using them to derive a noise ceiling is beyond the scope of this paper.
> > >
> > > We hope that this clarifies our comments on noise ceiling estimation: while it is possible to estimate a noise ceiling on single-trial data by treating data from other subjects as repeat trials, we believe that aligning data across subjects underestimates the noise ceiling for our analyses and diminishes the usefulness of such an estimate.

---

> > > > ### Comment · Reviewer_Ebp3 · 2023-08-16
> > > > **Thank you for the clarification**
> > > >
> > > > Dear authors,
> > > >
> > > > I appreciate the provided clarification regarding the estimation of the noise ceiling with single-trial data. I agree with the authors justification, and I am revising my score accordingly.
> > > >
> > > > However, I still have a lingering question that I would like to inquire about:
> > > >
> > > > Considering the insights from Schrimpf et al.'s 2020 paper, the authors computed the noise ceiling for the Pereira dataset involving the engagement of five subjects in reading passages. Despite the variance in the number of voxels across individual subjects, the authors proceeded to calculate the noise ceiling and subsequently reported achieving 100% normalized predictivity with the GPT2 model.
> > > >
> > > > I kindly request the authors to provide a clear explanation to shed light on this matter.

---

> > > > > ### Author Response · Authors · 2023-08-17
> > > > >
> > > > > Our understanding is that Schrimpf et al. used an ROI-based functional approach to compute noise ceilings for the Pereira et al. dataset (supplement line 96). Voxels were selected in each subject using a functional localizer (supplement line 82). While encoding models were evaluated for each voxel individually, it appears that the noise ceiling is computed using a population of voxels (supplement line 277). We therefore assume that correspondences between subjects were made at the level of localized ROIs rather than individual voxels, as we did not find any mention of anatomical or functional voxel-level alignment after the ROIs were localized.

---

### Official Review · Reviewer_1ukF · 2023-07-04

**Soundness:** 1 poor
**Presentation:** 2 fair
**Contribution:** 1 poor
**Rating:** 2
**Confidence:** 5

**Summary:**

This paper aims to fit fMRI data using transformer models. Of note, the paper aims to extrapolate from models that fit fMRI responses to visual stimuli to be able to account for fMRI responses to what may presumably be language stimuli. The results throughout the paper are extremely weak but are presented as if the models can provide insights into the fMRI signals.

**Strengths:**

The idea of comparing representations to visual stimuli and auditory stimuli is certainly very interesting.

The possibility of elucidating “conceptual” representations that transcend any one input modality is also quite interesting.


**Weaknesses:**

It would be great to start by demonstrating that the fMRI data actually relates to the visual and story stimuli. This should be shown by rigorous description of the fMRI data, with actual units, error bars, and statistics. There is nothing in the current paper that suggests that there is any explainable variance in the fMRI data.

Even though the paper does not show that the fMRI data contain any information, assuming that there is some degree of correlation between the fMRI data and the stimuli, and after all the undisclosed maneuvering, the results indicate that the models simply cannot capture the fMRI data. Take Fig. 1b. The uncorrected correlation coefficient hovers around 0. After more maneuvering and correction, the authors manage to get to 0.01. In other words, there is no correlation between the model outputs and the fMRI data.

Figure 3 is more challenging to interpret because the authors stop reporting r values and report normalized r values. This is a poor practice. For example, it could be that r_{movie > movie}=0.03 and r_{storie>movie}=0.01 and the ratio is 0.33 which might confuse the reader into thinking that there is something there. But the bottom line is that all of the r values are negligibly small.



**Questions:**

This paper would need an enormous amount of work to bring it to a level where it has a minimal scientific description of findings, starting with an open discussion of all the steps involved, documenting whether there is any reproducibility in the data, showing that the fMRI signals relate to language or auditory or visual stimuli, and finally potentially a role for neural network models. The current results leave it open as to whether the fMRI data relates to the stimuli and compellingly demonstrate that the neural networks cannot capture the fMRI data.

Other points.
Figure 4 includes enigmatic color scales that go from “-“ to “+”. Color scales should have numbers.


**Limitations:**

The authors failed to describe all the multiple limitations and to adequately interpret their results.

---

> ### Author Rebuttal · Authors · 2023-08-09
>
> Thank you for reviewing our paper and making these suggestions. We have addressed your concerns in order below:
>
> >It would be great to start by demonstrating that the fMRI data actually relates to the visual and story stimuli…
>
> We analyzed a publicly released dataset that has been used in previous studies relating fMRI responses to naturalistic stories and movies [1, 2, 3]. Furthermore, a large body of literature in the past two decades has demonstrated that fMRI responses to naturalistic stories and movies similar or identical to those used here contain meaningful information [2, 16, 17, 18], so a complete description of the fMRI dataset is out of the scope of this paper. In the final revision, we will emphasize that the fMRI data has been rigorously studied and modeled in prior peer-reviewed work.
>
> The fMRI response for a cortical voxel is given by the time-course of the blood-oxygen level dependent (BOLD) signal. The information in the fMRI response is contained not in the absolute value of the BOLD signal, but in its time-course. The objective of an encoding model is to predict the response time-course from stimulus features [19]. As described in Section 4.1, encoding models are evaluated by predicting brain responses to novel stimuli and computing the linear correlation between the predicted and the actual response time-courses. If we can successfully predict responses in a voxel from stimulus features, then our model has successfully explained some variance in the fMRI data.
>
> We report prediction performance in Figures 2, 3, and 5. Figure 2a shows that many voxels have cross-modality correlations greater than 0.1. Figure 3b shows that cross-modality correlations can reach 0.5 for $story→movie$ and 0.2 for $movie→story$, while within-modality correlations can reach 0.6 for $movie→movie$ and 0.4 for $story→story$. This demonstrates that there are many cortical regions where the encoding models explain a substantial fraction of the data variance.
>
> >Even though the paper does not show that the fMRI data contain any information…
>
> Figure 2b quantifies encoding model performance by taking the average correlation across all cortical voxels. As described in our reply to Reviewer C9Qg, averaging across cortex is an unbiased way to compare different encoding models. However, averaging across cortex does yield conservatively low correlation values, since there are many voxels (e.g. those in motor cortex) which are not involved in story or movie perception. Figure 2a shows that many regions involved in story or movie perception have correlations greater than 0.1, and Figure 3b shows that these correlations can reach up to 0.5 for $story→movie$ transfer, and up to 0.2 for $movie→story$ transfer. In the final paper, we will emphasize that the average correlation across cortex reported in Figure 2b is a conservatively low metric.
>
> >Figure 3 is more challenging to interpret because the authors stop reporting r values and report normalized r values…
>
> Please note that Figure 3a only shows $r_{source→target}/r_{target→target}$ for voxels with statistically significant (see Appendix A.1) values of $r_{target→target}$, so the ratios will not be inflated by negligibly small denominators. Furthermore, $r_{source→target}$ is directly compared to $r_{target→target}$ in the Figure 3b histograms, which show that many regions have high cross-modality and within-modality performance. Finally, Figure 3 is not meant to be a normalization of the results in Figure 2, but a way to compare cross-modality performance to within-modality performance. Many previous studies have demonstrated that language and vision encoding models achieve good within-modality performance (line 20), so within-modality performance provides a meaningful ceiling for cross-modality performance [4].
>
> In the final paper, we will emphasize that previous language and vision encoding models achieve good within-modality performance (line 20).
>
> >This paper would need an enormous amount of work to bring it to a level where it has a minimal scientific description of findings…
>
> Our study builds upon a commonly used framework for modeling brain responses to naturalistic stimuli using features extracted from neural networks [20]. There is little debate that fMRI signals relate to language and visual stimuli, as demonstrated by studies that read information about the stimulus out of fMRI data [10, 17, 18, 21, 22], even using the same dataset as that used here ([23]: “Decoding the semantic content…”). There is also little debate that neural network encoding models can successfully predict brain responses to a single stimulus modality after being trained on responses to that stimulus modality [11, 12, 13, 24, 25]. In the final paper, we will emphasize how our study builds on previous studies. We will also describe the fMRI dataset and the encoding model framework in more detail in the Appendices.
>
> The goal of our paper was to compare cross-modality performance to within-modality performance. We believe that our results convincingly demonstrate this, and that our analysis framework and the conclusions drawn from our results are comparable to previous fMRI modeling papers published in NeurIPS [11, 12].
>
> >Other points. Figure 4 includes enigmatic color scales that go from “-“ to “+”. Color scales should have numbers.
>
> Figure 4 shows the projection of encoding model weights onto principal components. We used “-” and “+” labels instead of numbers because the numerical scale of this projection is not meaningful.

---

> > ### Comment · Reviewer_1ukF · 2023-08-10
> > **Continued overinterpretation of fMRI signals**
> >
> > The authors rightly point out that there are many papers published with fMRI. There are also many papers published about extrasensorial perception. This does not make the work true.
> >
> > "The fMRI response for a cortical voxel is given by the time-course of the blood-oxygen level dependent (BOLD) signal." This is simply not true. Blood-oxygen level dependent refers to particular model that people often impose in this type of data. The authors are encouraged to report REAL data with REAL units.
> >
> > It has been quite common now to use neural network activations to fit signals from voxels. Unfortunately, it remains highly unclear whether this has led to any conceptual advancement. For example, it could be that an image makes people happy, that happiness leads to slightly more head movements, and this leads to some artifacts that can be correlated with the image. One can replace happiness with a lot of other variables, and head movements with a lot of other artifacts. Unfortunately, such data fitting does not help much, especially when dealing with signals that remain highly unclear and uncontrolled like those in fMRI.
> >
> > These concerns are especially notable when the fitting leads to such low performance as in the previous case. There are lots of indirect variables that could lead to the kind of small correlations reported here. It would be great to compare results with head movements, eye movements, attentional controls, and finally rigorous neurophysiological measurements.
> >
> > What does it mean from the point of view of neuroscience to "average across cortex"??? The voxels themselves are alreayd a massive averaging in spce and time with little meaning. This is an average of averages.
> >
> > In contrast to what the authors state, there is extensive debate about whether fMRI responses relate to language or vision, except within the fMRI community of course. It would be very useful to verify any of the claims with real neurophysiological data. Citing fMRI data to justify fMRI data is circular. What is needed is rigorous independent validation.

---

> > > ### Author Response · Authors · 2023-08-21
> > >
> > > These concerns are addressed at the field of fMRI research, rather than the specifics of our current work. They are also contrary to the general consensus in the field of neuroscience. However, we appreciate that many readers may be unfamiliar with fMRI and neuroscience methods generally, so we will provide a brief context here.
> > >
> > > First, many groups of researchers have independently established the correspondence between fMRI and the physiological processes in the brain, as well as between fMRI and other neurophysiological measures. This is a large body of literature spanning research across physics, biology, and neuroscience. Here are some representative examples:
> > >
> > > - [1] and [2] both provide great literature reviews on the reasons and evidence showing why fMRI signals are dependent on the blood oxygen metabolism. Note that in as early as 1992, three independent groups were able to relate fMRI signals in human brains to task stimuli [3,4,5].
> > > - Many additional studies (such as [6,7,8,9,10,11,12]) have validated fMRI responses by establishing a direct link between fMRI signals and the local neuronal activities measured by microelectrode recordings. This was shown in different animals (monkeys, cats, rats) and humans.
> > > - Other studies (such as [13,14,15]) compared fMRI signals in human brains to ECoG and/or EEG data, and again found reliable links between these measures.
> > > - Research has also shown that human cortical activities are reliable both within and across subjects in response to naturalistic stimuli, such as those used in our study (see [16] for a review). Again, this was demonstrated by ECoG and EEG data, in addition to fMRI data. The 3 measures are correlated [15].
> > >
> > > Second, studies that related representations in deep neural networks and human fMRI signals have also been replicated using other measures. For example:
> > > - [17] found converging results using fMRI and MEG.
> > > - [18] found converging results using fMRI and ECoG.
> > >
> > > In conclusion, there is rigorous evidence across decades of peer-reviewed literature showing independent validation of fMRI data, using various methods other than fMRI. This has led the overwhelming majority of neuroscientists to agree that the BOLD signal, as measured in the dataset we report, is a meaningful measure of brain activity that corresponds to internal representations and experiences of the model organism. We also note that these analyses are routinely corrected for physiological confounds such as head movements, breathing, etc. This is also the case for the dataset we used.
> > >
> > >
> > > [1] Logothetis. 2003. The underpinnings of the BOLD functional magnetic resonance imaging signal. Journal of Neuroscience
> > >
> > > [2] Kim & Ogawa. 2012. Biophysical and physiological origins of blood oxygenation level-dependent fMRI signals. Journal of Cerebral Blood Flow & Metabolism
> > >
> > > [3] Bandettini, … 1992. Time course EPI of human brain function during task activation. Magnetic Resonance in Medicine
> > >
> > > [4] Kwong, … 1992. Dynamic magnetic resonance imaging of human brain activity during primary sensory stimulation. Proceedings of the National Academy of Sciences (PNAS)
> > >
> > > [5] Ogawa, … 1992. Intrinsic signal changes accompanying sensory stimulation: functional brain mapping with magnetic resonance imaging. PNAS
> > >
> > > [6] Logothetis, … 2001. Neurophysiological investigation of the basis of the fMRI signal. Nature
> > >
> > > [7] Niessing, … 2005. Hemodynamic signals correlate tightly with synchronized gamma oscillations. Science
> > >
> > > [8] Mukamel, … 2005. Coupling between neuronal firing, field potentials, and fMRI in human auditory cortex. Science
> > >
> > > [9] Smith, … 2002. Cerebral energetics and spiking frequency: the neurophysiological basis of fMRI. PNAS
> > > [10] Logothetis. 2002. The neural basis of the blood-oxygen-level-dependent functional magnetic resonance imaging signal. Philosophical Transactions of the Royal Society of London
> > >
> > > [11] Shmuel, … 2006. Negative functional MRI response correlates with decreases in neuronal activity in monkey visual area V1. Nature Neuroscience
> > >
> > > [12] Nir, … 2007. Coupling between neuronal firing rate, gamma LFP, and BOLD fMRI is related to interneuronal correlations. Current Biology
> > >
> > > [13] Debener, … 2005. Trial-by-trial coupling of concurrent electroencephalogram and functional magnetic resonance imaging identifies the dynamics of performance monitoring. Journal of Neuroscience
> > >
> > > [14] Hermes, … 2012. Neurophysiologic correlates of fMRI in human motor cortex. Human brain mapping
> > >
> > > [15] Haufe, … 2018. Elucidating relations between fMRI, ECoG, and EEG through a common natural stimulus. NeuroImage
> > >
> > > [16] Hasson, … 2010. Reliability of cortical activity during natural stimulation. Trends in Cognitive Sciences
> > >
> > > [17] Cichy, … 2016. Comparison of deep neural networks to spatio-temporal cortical dynamics of human visual object recognition reveals hierarchical correspondence. Scientific Reports
> > >
> > > [18] Schrimpf, ... 2021. The neural architecture of language: Integrative modeling converges on predictive processing. PNAS

---

### Official Review · Reviewer_LQBp · 2023-07-06

**Soundness:** 2 fair
**Presentation:** 2 fair
**Contribution:** 2 fair
**Rating:** 7
**Confidence:** 3

**Summary:**

This work seeks to identify multi-modal processing in the brain. The proposed technique can be summarized as follows: a language encoding model is trained to predict neural story responses from story representations, and a vision language encoding model is used to predict movie responses from movie representations. Then, the language encoding model is used to try to predict movie responses from movie representations. The quality of predictions is used to measure the amount of cross-modal information being processed in a given brain region. An analogous procedure is carried out for the vision encoding model.

**Strengths:**

- As far as I can tell, the technique is novel. There are other works that look for multi-modality in the brain by trying to map multi-modal model activation to neural activation, but this is the first work to look for shared conceptual representation in the brain directly, by training two separate vision and language encoders and comparing the prediction performances after swapping the modalities.
- This method seems mostly sound, with some remaining areas that could still be clarified (see weaknesses)

**Weaknesses:**

- The experiment described in section 5.1 and figure 2 is difficult to interpret on its own. The ability to predict visual response from a model trained on language stimuli is not enough to conclude that an area is multimodal. There is some uncertainty due to the fact that we don't know the quality of the alignment between feature spaces (discussed in section 2.2). In the limit, if the alignment between features is perfect, then the representation of an image of a cat can be perfectly aligned to the representation of the word "cat", and from there it becomes unclear whether a good $r_{movie\rightarrow story}$ is due to a good alignment or true multi-modality in the brain.
- Fortunately, this problem is partially addressed by normalizing the scores by within-modality performance, as is done in 5.2, but even then, a poor alignment could result in poor correlations across all regions. For both cases, it would be helpful to see the quality of the alignment, perhaps in the appendix.
- I had trouble understanding the methods discussed in section 5.3 "Encoding model principal components" (see questions below).
- My biggest concern with this work is significance to the NeurIPS community. The most interesting results, the identification of cross-modal areas in the brain, seem to be neuroscience related. The most relevant ML takeaways from this work are already well known, namely the benefits of multi-modal training.

**Questions:**

- line 129: is it possible to be more specific about what the movie stimuli were like? Did the movies have dialogue? Music?
- Figure 2: The white lines denoting the ROIs are hard to see. How exactly are these ROIs obtained?
- Figure 2: It would also be helpful to label the visual area and the auditory areas on the inset 3D brains as well.
- Figure 3: For $r_{story\rightarrow movie}/r_{movie\rightarrow movie}$, is the upper bound of this quantity 1 by default? Or did it just so happen that none of the cross-modal scores were better than the within-modal scores
- Line 215-223 discuss reasons why the $r_{movie\rightarrow story}$ were lower than $r_{story \rightarrow movie}$. Is there a way to rule out the possibility that the alignment of feature spaces (discussed in 2.2) is simply worse in one direction?
- Line 229: By what metric are the "top" voxels selected?
- Line 231: How can stimulus features be projected onto each PC? Don't the stimulus features and the encoding weights occupy completely different spaces?

**Limitations:**

Limitations and negative societal impact are not discussed, but are probably not relevant here.

---

> ### Author Rebuttal · Authors · 2023-08-09
>
> Thank you for reviewing our paper and making these suggestions. We have addressed your concerns in order below:
>
> >The experiment described in section 5.1 and figure 2 is difficult to interpret on its own…
>
> We respectfully disagree with this assessment. Take the suggested example where the BridgeTower representation for cats in vision is identical to the BridgeTower representation for the word "cat" in language, and we are evaluating if a voxel that responds to cats in vision is multimodal (i.e. if it responds similarly to the word “cat” in language). During training, the encoding model learns that the voxel has a response of $β_{cat}$ when cats appear in movies. During $movie→story$ transfer, the encoding model predicts a response of $β_{cat}$ when the word “cat” occurs in stories. Transfer performance will only be high if the voxel has an actual response of $β_{cat}$ to the word “cat” in language.
>
> In other words, a region with good transfer performance must respond similarly to concepts in language and vision, which is how we operationalize multimodality in the brain (line 22). In the final paper, we will emphasize how our transfer metric $r_{source→target}$ (line 150) relates to our definition of multimodality (line 22).
>
> >Fortunately, this problem is partially addressed by normalizing the scores by within-modality performance…
>
> Thank you for raising this point. We will include our measure of alignment quality in the final paper.
>
> We estimated alignment matrices using the Flickr30k dataset (line 103). To test alignment quality, we estimated alignment matrices on 80% of the image-caption pairs and evaluated them on the remaining 20%. We scored alignment quality by correlating the predicted and actual values for each feature across test pairs, and averaging the correlations across features. Finally, we normalized alignment scores by reporting the number of standard deviations above the mean of a null distribution, which we obtained by randomly shuffling the test captions and images.
>
> These results are shown in Figure R1. The $caption→image$ matrices had a normalized score of 221.30 σ, while the $image→caption$ matrices had a normalized score of 197.28 σ. Thus in both directions the alignment was roughly 200 standard deviations better than random.
>
> >I had trouble understanding the methods discussed in section 5.3…
>
> The methods used in Section 5.3 were based on previous studies that used PCA to characterize semantic tuning in the brain [1, 2, 4, 5]. We will clarify these methods in Appendix A.2 of the final paper.
>
> >My biggest concern with this work is significance to the NeurIPS community…
>
> Many papers recently published in NeurIPS [11, 12, 15] have shown that understanding the relationships between human and artificial information processing systems can benefit both fields of study. We believe that our results relating multimodal transformers and multimodal representations in the brain will be of interest to neuroscientists, computer scientists, and the growing number of researchers who work in the intersection of the fields.
>
> >line 129: is it possible to be more specific about what the movie stimuli were like…
>
> The movie stimuli (https://gin.g-node.org/gallantlab/shortclips/src/master/stimuli) originally had dialogue and music, but the clips were presented silently to the subjects. We will clarify this in the final paper.
>
> >Figure 2: The white lines denoting the ROIs are hard to see…
>
> ROIs were identified using functional localizers described in [1]. The relevant ROIs for this paper are auditory cortex (AC; identified using segments of music, speech, and natural sounds) and primary visual cortex (V1; identified using rotating wedges and expanding / contracting rings). In the final paper we will make the ROI lines more visible and include details about ROI localization in the Appendix.
>
> >Figure 2: It would also be helpful to label the visual area and the auditory areas on the inset 3D brains as well.
>
> We will label the 3D brains in the final paper.
>
> >Figure 3: For r_story→movie/r_movie→movie, is the upper bound of this quantity 1 by default…
>
> $r_{s→m}/r_{m→m}$ does not have a strict upper bound at 1. For instance, $r_{s→m}$ may be higher than $r_{m→m}$ if the story and movie datasets differ in semantic coverage (line 217) or signal-to-noise ratio. Nonetheless, $r_{s→m}/r_{m→m}$ rarely exceeded 1 in voxels that were well-predicted by the vision encoding model (Figure 3).
>
> >Line 215-223 discuss reasons why the r_movie→story were lower than r_story→movie…
>
> As mentioned above, we tested the quality of the alignment matrices by holding out test pairs from the Flickr30k dataset. Alignment was much higher than chance level and fairly symmetric, so it is unlikely that the asymmetry observed in Figure 3 was due to differences in alignment quality.
>
> Previous studies have observed a similar asymmetry [15], and we agree that characterizing the sources of this asymmetry is an important direction for future work.
>
> >Line 229: By what metric are the "top" voxels selected?
>
> We selected voxels using a cross-validation procedure (line 143). In each iteration, 20% of the timepoints were removed from the training dataset and reserved for validation. Models were estimated on the remaining 80% of the timepoints and used to predict responses for the reserved timepoints. The 10,000 voxels with the highest linear correlations were used for PCA. We will clarify this in the final paper.
>
> >Line 231: How can stimulus features be projected onto each PC…
>
> Encoding model weights predict how a voxel responds to stimulus features. For instance, a voxel that responds to animal words will have weights that are aligned with the BridgeTower features of animal words. The only difference is that encoding model weights have 4 delays for each feature (line 137). We remove this temporal information from the weights before PCA by averaging across the four delays for each feature [2]. We will clarify this in the final paper.

---

> > ### Comment · Reviewer_LQBp · 2023-08-21
> > **Response to authors rebuttal**
> >
> > I thank the authors for their thorough response.
> > > Re:  experiment described in section 5.1 and figure 2
> > I can at least believe that good transfer performance can only be due to multimodality. But I still believe that bad performance could also be due to poor (but out of the author's control) alignment between vision and language feature spaces. When it comes to identifying multimodal ROIs, I think this could lead to some false negatives. But I don't think this too great a weakness.
> >
> > Otherwise, I have been convinced by this work's relevance to the NeurIPs community, and will increase the score to 7.

---

### Official Review · Reviewer_V1MV · 2023-07-07

**Soundness:** 3 good
**Presentation:** 4 excellent
**Contribution:** 3 good
**Rating:** 7
**Confidence:** 4

**Summary:**

In this paper, the authors used a pre-trained multimodal transformer, the BridgeTower model, to obtain feature representations of language and visual stimuli in the fMRI experiments. They trained encoding models on these representations and brain responses, and demonstrated that models trained on one modality could predict brain responses to another modality. Brain regions with aligned representations and semantic dimensions were discovered. The authors also defined a correlation score for evaluation and performed comparison with unimodal transformers.

**Strengths:**

Investigating the multimodal processing of brain activity is an interesting problem. This paper presented a novel idea to develop brain encoding models from representations of different stimulus modality learned from a multimodal transformer and fMRI responses. It provides a new approach for analyzing the alignment of language and visual representations in the brain.

**Weaknesses:**

The authors used a publicly available dataset containing fMRI responses from only five subjects, which is quite limited. Some of the discoveries, including brain regions for representation alignment, semantic dimensions, and the differences in cross-modality process, need to be tested or validated with experiment data.

**Questions:**

Besides the score defined from correlation, what are some other possible measurements? What are the inter-subject differences?

**Limitations:**

The limitations were not adequately addressed in the paper. One possible discussion could be about the data limitation, and future work should include more subjects, maybe from different cultural backgrounds speaking different languages.

---

> ### Author Rebuttal · Authors · 2023-08-09
>
> Thank you for reviewing our paper and making these suggestions. We have addressed your concerns in order below:
>
> >The authors used a publicly available dataset containing fMRI responses from only five subjects, which is quite limited. Some of the discoveries, including brain regions for representation alignment, semantic dimensions, and the differences in cross-modality process, need to be tested or validated with experiment data.
>
> Although five subjects is low relative to some other human neuroimaging studies, this number is comparable to previous studies that model brain responses to naturalistic language [2, 6] and vision [1, 7, 8]. In many experiments involving naturalistic stimuli, a small number of subjects is scanned, but a large amount of brain data is collected for each subject. This allows for the replication of the effects in each individual subject (Appendix C), which can be more powerful than looking for average effects across a group [9]. However, we agree that having a larger and more diverse dataset is preferable, and we will discuss this limitation in the final paper.
>
> All of the results of the study were based on data from fMRI experiments, so we would appreciate it if the reviewer could further clarify what it means to “test or validate them with experiment data”.
>
> >Besides the score defined from correlation, what are some other possible measurements [for encoding model performance]?
>
> Some studies quantify encoding model prediction performance using a classification task, in which encoding model predictions are used to predict which of two stimulus segments occurred at a given time [6, 10, 11]. However, linear correlation is the most commonly used metric used to quantify encoding model prediction performance [2, 12, 13, 14]. We will reference the studies that use this metric in the final paper.
>
> >What are the inter-subject differences [for encoding model performance]?
>
> Our flatmaps suggest that the effects that we observe are consistent across subjects (Appendix C). In Figures 2 and 4, we summarize variability across subjects using error bars. We will note how our results are similar and different across subjects in the final paper.
>
> >The limitations were not adequately addressed in the paper. One possible discussion could be about the data limitation, and future work should include more subjects, maybe from different cultural backgrounds speaking different languages.
>
> Thank you for raising this point. In the final paper, we will add a third discussion paragraph summarizing limitations, including the stimulus limitations discussed in Section 5.2 (line 217) and the dataset limitations discussed above.

---

> > ### Comment · Reviewer_V1MV · 2023-08-17
> > **Thanks for rebuttal**
> >
> > Dear authors,
> >
> > Thanks for your rebuttal!
> >
> > I appreciate the effort in clarifying and addressing my concerns, adding discussion about limitations. For the first point “test or validate them with experiment data”, I mean that many discoveries made in this paper were derived from the limited fMRI dataset of five subjects. It would be better if the authors could provide more references or experimental results (if any) to support these discoveries.
> >
> > Overall, I see the novelty and contribution of this work, and would still recommend "Accept".

---

### Official Review · Reviewer_C9Qg · 2023-07-07

**Soundness:** 3 good
**Presentation:** 3 good
**Contribution:** 3 good
**Rating:** 7
**Confidence:** 4

**Summary:**

This paper studies whether we can use multi-modal transformers to predict brain activities. It specifically uses neural nets’ encoding for languages to predict brains’ activities for seeing vidieos, and neural nets’ encoding for images to predict brains’ activities. Their finding is that these prediction models are statistically significant so multi-modal transformers may resemble the underlying mechanism of brains.


**Strengths:**


Understanding how neural nets are related to real brains has been a central question in deep learning. So this paper makes an important addition to the literature. While prior works already used similar frameworks (using encoding for transformers to predict MRI data/images), this work still seems quite impressive because it needs to circumvent quite a few kinks, such as aligning the visual and text encoding and carefully designing the statistical experiments to confirm models significance because the r2/r are fairly small.


**Weaknesses:**

I feel the paper has a very high bar on the audience, i.e., requiring one to have sufficient knowledge on transformers, MRI, and statistical analysis (multiple hypothesis test). I don’t know what to suggest to fix the problem as the paper is already very carefully written. Maybe having Appendices A and B to give more background materials on MRI and the statistical tools they use.

**Questions:**



1.	How can one make sense of the very weak correlation results? Like there are some statistical evidence that transformers resemble the brain but transformers are only < 1% like brains?
2.	Do those t(4) mean t-stat/z-score? Do we need any kind of serial adjustments when one calculate these scores because I imagine input and/or response could potentially be correlated. I guess some bootstrapping machinaries are designed to address/alleviate this issue?
3.	Why the correlation numbers in Fig 3 & 5 are much higher than those in Fig. 2? I guess I completely missed a fraction of the paper…
4.	Do it still make sense to do PCA analysis when a model’s r2 is already that low? Some t-stat seems to drop to 2-ish (so shall we interpret them as significant or not?).

---

> ### Author Rebuttal · Authors · 2023-08-09
>
> Thank you for reviewing our paper and making these suggestions. We have addressed your concerns in order below:
>
> >I feel the paper has a very high bar on the audience, i.e., requiring one to have sufficient knowledge on transformers, MRI, and statistical analysis (multiple hypothesis test). I don’t know what to suggest to fix the problem as the paper is already very carefully written. Maybe having Appendices A and B to give more background materials on MRI and the statistical tools they use.
>
> We used an fMRI dataset and an encoding model framework that have been described in several previous studies [1, 2, 3]. As a result, we focused our methods sections on explaining the aspects of our study that differ from previous studies. In the final paper, we will emphasize how our study builds on previous studies. We will also describe the fMRI dataset and the encoding model framework in more detail in Appendices A and B.
>
> >How can one make sense of the very weak correlation results? Like there are some statistical evidence that transformers resemble the brain but transformers are only < 1% like brains?
>
> Figure 2b quantifies encoding model performance by taking the average correlation across all cortical voxels. Many fMRI studies select a subpopulation of voxels with an independent localizer task that isolates the cognitive effect of interest. However, we do not believe there is an effective multimodal localizer that would select all voxels that would represent the semantic dimensions of interest. Averaging across cortex is therefore an unbiased way to compare the different encoding models. However, averaging across cortex does yield conservatively low correlation values, since there are many voxels (e.g. those in motor cortex) which are not involved in story or movie perception. Figure 2a shows that many regions involved in story or movie perception have correlations greater than 0.1, and Figure 3b shows that these correlations can reach 0.5 for $story→movie$ transfer and 0.2 for $movie→story$ transfer. In the final paper, we will emphasize that Figure 2b reports average correlation across cortex, which is a conservative but unbiased metric.
>
> We will also note that we evaluated encoding models on single-trial fMRI responses, whereas many previous studies evaluated encoding models on fMRI responses averaged across multiple repeats of the same stimulus. Evaluating on single-trial responses to a large number of stimuli provides high semantic coverage. However, averaging across repeats will increase the signal-to-noise ratio of the data and lead to higher correlations than those reported in our study.
>
> Thank you for bringing up this point, and we hope that clarifying the correlation results will strengthen our paper.
>
> >Do those t(4) mean t-stat/z-score? Do we need any kind of serial adjustments when one calculate these scores because I imagine input and/or response could potentially be correlated. I guess some bootstrapping machinaries are designed to address/alleviate this issue?
>
> Yes, $t$(4) indicates a t-test with 4 degrees of freedom. These tests were conducted across 5 individuals, so there is no autocorrelation that needs to be accounted for.
>
> In analyses where we computed significance within each voxel (e.g. to obtain the voxel masks used in Figure 3) we used a blockwise permutation test to account for the autocorrelation in the responses (Appendix A.1). We will describe this process in more detail (line 158) in the final paper.
>
> >Why the correlation numbers in Fig 3 & 5 are much higher than those in Fig. 2? I guess I completely missed a fraction of the paper…
>
> As mentioned above, Figure 2b reports average correlation across cortex, while the histograms in Figures 3b and 5b report correlations for individual cortical locations. Further, the histograms in Figures 3b and 5b are restricted to voxels with statistically significant $target→target$ scores. In the final paper, we will emphasize that Figure 2b reports average correlation across cortex.
>
> >Do it still make sense to do PCA analysis when a model’s r2 is already that low? Some t-stat seems to drop to 2-ish (so shall we interpret them as significant or not?).
>
> We performed PCA using the top 10,000 voxels for each subject, which have an average correlation of 0.045. This approach is consistent with what has been done in previous studies [1, 2, 4, 5]. Furthermore, we found principal components that are consistent with those found in these previous studies.
>
> All of the reported t-tests were statistically significant after correcting for multiple comparisons. The different t-tests correspond to different analyses (the first test compares multimodal transfer performance before and after correcting for negative correlations, whereas the second test compares multimodal transfer performance to unimodal transfer performance), so the t-statistics are not comparable across tests.

---

### Author Rebuttal · Authors · 2023-08-09

We thank the reviewers for taking the time to carefully read our paper, and for providing detailed feedback. We are excited to see that many reviewers agree that our approach is novel and yields interesting insights, and we sincerely appreciate the concerns and suggestions raised by all reviewers. In response, we have addressed these points in the subsequent sections for each individual reviewer, and will incorporate changes accordingly in the final paper. We hope these will serve to amplify the clarity and robustness of our research.

We have attached a list of figures, which we reference in the rebuttals as R1, R2, R3, and R4.

We reference the following papers in the rebuttals:

[1] Huth, A. G., Nishimoto, S., Vu, A. T. & Gallant, J. L. A continuous semantic space describes the representation of thousands of object and action categories across the human brain. Neuron 76, 1210–1224 (2012).

[2] Huth, A. G., de Heer, W. A., Griffiths, T. L., Theunissen, F. E. & Gallant, J. L. Natural speech reveals the semantic maps that tile human cerebral cortex. Nature 532, 453–458 (2016).

[3] Popham, S. F. et al. Visual and linguistic semantic representations are aligned at the border of human visual cortex. Nat. Neurosci. 24, 1628–1636 (2021).

[4] Deniz, F., Nunez-Elizalde, A. O., Huth, A. G. & Gallant, J. L. The Representation of Semantic Information Across Human Cerebral Cortex During Listening Versus Reading Is Invariant to Stimulus Modality. J. Neurosci. 39, 7722–7736 (2019).

[5] Wang, A. Y., Kay, K., Naselaris, T., Tarr, M. J. & Wehbe, L. Incorporating natural language into vision models improves prediction and understanding of higher visual cortex. bioRxiv 2022.09.27.508760 (2022) doi:10.1101/2022.09.27.508760.

[6] Wehbe, L. et al. Simultaneously uncovering the patterns of brain regions involved in different story reading subprocesses. PLoS One 9, e112575 (2014).

[7] Shen, G., Horikawa, T., Majima, K. & Kamitani, Y. Deep image reconstruction from human brain activity. PLoS Comput. Biol. 15, e1006633 (2019).

[8] Allen, E. J. et al. A massive 7T fMRI dataset to bridge cognitive neuroscience and artificial intelligence. Nat. Neurosci. 25, 116–126 (2022).

[9] Braga, R. M. & Buckner, R. L. Parallel Interdigitated Distributed Networks within the Individual Estimated by Intrinsic Functional Connectivity. Neuron 95, 457–471.e5 (2017).

[10] Mitchell, T. M. et al. Predicting human brain activity associated with the meanings of nouns. Science 320, 1191–1195 (2008).

[11] Toneva, M. & Wehbe, L. Interpreting and improving natural-language processing (in machines) with natural language-processing (in the brain). in Advances in Neural Information Processing Systems (2019).

[12] Jain, S. & Huth, A. Incorporating Context into Language Encoding Models for fMRI. in Advances in Neural Information Processing Systems (2018).

[13] Schrimpf, M. et al. The neural architecture of language: Integrative modeling converges on predictive processing. Proc. Natl. Acad. Sci. U. S. A. 118, e2105646118 (2021).

[14] Caucheteux, C. & King, J.-R. Brains and algorithms partially converge in natural language processing. Commun. Biol. 5, 134 (2022).

[15] Lin, S., Sprague, T., & Singh, A. K. Mind reader: Reconstructing complex images from brain activities. in Advances in Neural Information Processing Systems (2022).

[16] Hasson, U., Malach, R. & Heeger, D. J. Reliability of cortical activity during natural stimulation. Trends Cogn. Sci. 14, 40–48 (2010).

[17] Nishimoto, S. et al. Reconstructing visual experiences from brain activity evoked by natural movies. Curr. Biol. 21, 1641–1646 (2011).

[18] Tang, J., LeBel, A., Jain, S. & Huth, A. G. Semantic reconstruction of continuous language from non-invasive brain recordings. Nat. Neurosci. 26, 858–866 (2023).

[19] Naselaris, T., Kay, K. N., Nishimoto, S. & Gallant, J. L. Encoding and decoding in fMRI. Neuroimage 56, 400–410 (2011).

[20] Jain, S., Vo, V. A., Wehbe, L. & Huth, A. G. Computational language modeling and the promise of in silico experimentation. Neurobiol. Lang. (Camb.) doi:10.1162/nol_a_00101/114613.

[21] Haxby, J. V. et al. Distributed and overlapping representations of faces and objects in ventral temporal cortex. Science 293, 2425–2430 (2001).

[22] Kay, K. N., Naselaris, T., Prenger, R. J. & Gallant, J. L. Identifying natural images from human brain activity. Nature 452, 352–355 (2008).

[23] Huth, A. G. et al. Decoding the Semantic Content of Natural Movies from Human Brain Activity. Front. Syst. Neurosci. 10, 81 (2016).

[24] Yamins, D. L. K. et al. Performance-optimized hierarchical models predict neural responses in higher visual cortex. Proc. Natl. Acad. Sci. U. S. A. 111, 8619–8624 (2014).

[25] Güçlü, U. & van Gerven, M. A. J. Deep neural networks reveal a gradient in the complexity of neural representations across the ventral stream. Journal of Neuroscience 35, 10005–10014 (2015).

[26] Liu, Y. et al. KD-VLP: Improving End-to-End Vision-and-Language Pretraining with Object Knowledge Distillation. arXiv [cs.CV] (2021).

---

### Decision · Program_Chairs · 2023-09-21

**Decision:**

Accept (poster)

**Comment:**

The manuscript demonstrate cross-modality decoding in fMRI where a language model predicts visual responses and a vision model predicts language responses. Authors find aligned visual-language representations in many areas of the brain. Reviewers, bar one who had fundamental concerns about fMRI itself, were unanimous in their recommendation that the manuscript makes an important contribution.

Reviewers appreciated the fairly simple method whose results are easy to interpret. While the impact on both neuroscience and ML is fairly limited at present, finding multimodal transformer architectures that are more brain-like could have high impact in the future, as could understanding how multimodal information is processed in the brain. This work is a prerequisite for both of these directions.

Reviewers repeatedly asked for a thorough discussion about limitations. I would have liked to read this before acceptance, but this year manuscripts could not be updated and unfortunately reviewers were very limited in the response they could provide to authors. Authors promised such a section and I encourage them to follow through, as the reviewers mentioned multiple possible limitations which are necessary for contextualizing this work and for enabling followup research.